# WEIGHT SPACE REPRESENTATION LEARNING ON DIVERSE NeRF ARCHITECTURES

**Francesco Ballerini, Pierluigi Zama Ramirez, Luigi Di Stefano, Samuele Salti**
University of Bologna, Italy
`francesco.ballerini4@unibo.it`

## ABSTRACT

Neural Radiance Fields (NeRFs) have emerged as a groundbreaking paradigm for representing 3D objects and scenes by encoding shape and appearance information into the weights of a neural network. Recent studies have demonstrated that these weights can be used as input for frameworks designed to address deep learning tasks; however, such frameworks require NeRFs to adhere to a specific, predefined architecture. In this paper, we introduce the first framework capable of processing NeRFs with diverse architectures and performing inference on architectures unseen at training time. We achieve this by training a Graph Meta-Network within an unsupervised representation learning framework, and show that a contrastive objective is conducive to obtaining an architecture-agnostic latent space. In experiments conducted across 13 NeRF architectures belonging to three families (MLPs, tri-planes, and, for the first time, hash tables), our approach demonstrates robust performance in classification, retrieval, and language tasks involving multiple architectures, even unseen at training time, while also matching or exceeding the results of existing frameworks limited to single architectures. Our code and data are available at this https URL.

## 1    INTRODUCTION

Neural Radiance Fields (NeRFs) (Mildenhall et al., 2020) have emerged over the last few years as a new paradigm for representing 3D objects and scenes (Xie et al., 2022). A NeRF is a neural network trained on a collection of images to map 3D coordinates to color and density values, which can then be used to synthesize novel views of the underlying object or scene via volume rendering. Due to their continuous nature, NeRFs can encode an arbitrary number of images at any resolution into a finite number of neural network weights, thus decoupling the number of observations and their spatial resolution from the memory required to store the 3D representation. As a result, NeRFs hold the potential to become a standard tool for storing and communicating 3D information, as supported by the recent publication of several NeRF datasets (De Luigi et al., 2023; Hu et al., 2023; Zama Ramirez et al., 2024; Cardace et al., 2024; Amaduzzi et al., 2025).

With the rise of NeRFs as a new data format, whether and how it is possible to perform traditional deep learning tasks on them has become an increasingly relevant research question. The naive solution to this problem involves rendering views of the underlying object from its NeRF representation and leveraging existing neural architectures designed to process images. However, this procedure requires additional computation time and several decisions that are likely to impact its outcome, such as the number of views to render, their viewpoint, and their resolution. A more elegant and efficient approach, explored in recent works, relies on performing tasks on NeRFs by processing their weights as input, thereby requiring no rendering step. This strategy is adopted by `nf2vec` (Zama Ramirez et al., 2024), a representation learning framework that learns to map NeRF weights to latent vectors by minimizing a rendering loss, where NeRFs are standard Multi-Layer Perceptrons (MLPs). These vectors are then used as input to deep learning pipelines for downstream tasks. A related approach is proposed by Cardace et al. (2024), who, instead of employing traditional MLPs, leverage tri-planar NeRFs (Chan et al., 2022), where input coordinates are projected onto three orthogonal planes of learnable features to compute the input for an MLP.

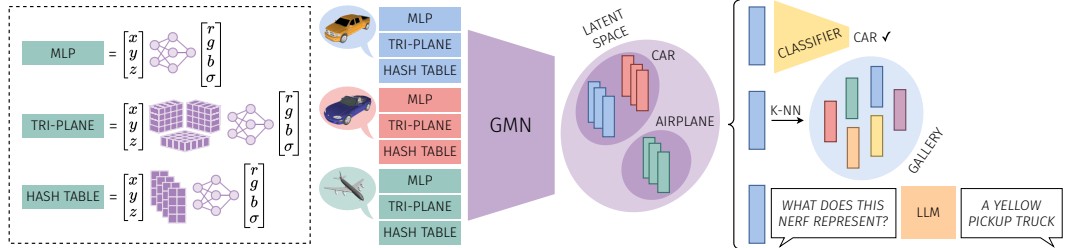

Figure 1: **Framework overview.** Our representation learning framework leverages a Graph Meta-Network (Lim et al., 2024) encoder to map weights of NeRFs with diverse architectures to a latent space where NeRFs representing similar objects are close to each other, regardless of their architecture. The embeddings are then used as input to downstream pipelines for classification, retrieval, and language tasks.

Both nf2vec (Zama Ramirez et al., 2024) and the method by Cardace et al. (2024), however, are designed to ingest a specific type of NeRF architecture (i.e. MLPs with fixed hidden dimensions in nf2vec and tri-planes with fixed spatial resolution in Cardace et al.), and are thus unable to process diverse input architectures. In the context of a research domain where new NeRF designs are constantly being explored (Xie et al., 2022), this assumption strongly limits their applicability. The issue of handling arbitrary architectures has recently been studied in the broader research field on *meta-networks*, i.e. neural networks that process other neural networks as input. Specifically, Graph Meta-Networks (GMNs) have been proposed (Lim et al., 2024; Kofinas et al., 2024), namely Graph Neural Networks (GNNs) that can ingest any neural architecture as long as it can be first converted into a graph. Yet, these works do not experiment with NeRFs as input to GMNs.

Motivated by the potential of GMNs to process diverse NeRF architectures, we investigate whether a GMN encoder can learn a latent space where distances reflect the similarity between the actual content of the radiance fields rather than their specific neural parameterization. Our empirical study reveals that this latent space organization cannot be achieved by solely relying on a rendering loss (Zama Ramirez et al., 2024), as such a loss alone causes different NeRF architectures to aggregate into distinct clusters in the embedding space, even when they represent the same underlying object. To overcome this limitation, we draw inspiration from the contrastive learning literature and introduce a SigLIP loss term (Zhai et al., 2023) that places pairs of NeRFs with different architectures representing the same object close to each other in latent space while pushing other pairs further apart. Combined with a rendering loss, this approach enforces an architecture-agnostic embedding space organized by class and instance. Our experimental study features three families of popular NeRF architectures, namely MLPs, tri-planes (Chan et al., 2022), and multi-resolution hash tables (Müller et al., 2022), for a total of 13 diverse architectures, and reveals that our encoder produces latent representations that serve as effective inputs for downstream tasks such as classification, retrieval, and language tasks. Notably, our framework is the first to perform tasks on NeRF parameterized as hash tables by processing their weights. Fig. 1 outlines the key components of our framework.

Our contributions can be summarized as follows:

- We present the first framework that performs tasks on NeRFs parameterized by diverse architectures by processing their weights.
- We propose to use a contrastive learning objective to create a latent space where NeRFs with similar content are close to each other, regardless of their architecture.
- We tackle, for the first time, downstream tasks on NeRFs parameterized by hash tables.
- We show that, within the families seen at training time, our framework can effectively process unseen NeRF architectures and generalize to NeRFs trained on unseen data.
- We achieve comparable or superior results to previous methods operating on single architectures.

## 2 RELATED WORK

**Neural Radiance Fields.** NeRFs were first introduced by Mildenhall et al. (2020) as a method for novel view synthesis, namely the task of generating previously unseen views of a scene from a

set of sparse input images taken from different viewpoints. In the original formulation, a NeRF is an MLP that parameterizes a function $(x, y, z, \theta, \phi) \mapsto (r, g, b, \sigma)$ that maps a 3D position $(x, y, z)$ and a 2D viewing direction $(\theta, \phi)$ to an emitted color $(r, g, b)$ and volume density $\sigma$. Since then, several architectural variants have been proposed, many of which combine the MLP with a trainable discrete data structure that quantizes the space of input coordinates and maps them to a higher-dimensional vector, which then serves as the actual MLP input. These structures include voxel grids (Liu et al., 2020), tri-planes (Chan et al., 2022), and multi-resolution hash tables (Müller et al., 2022), and typically result in NeRF architectures that can be trained much faster to convergence without sacrificing rendering quality. In particular, hash tables are the most widely adopted NeRF architecture in recent works (Barron et al., 2023; Wang et al., 2024; Hu et al., 2024; Lee et al., 2025; Chen et al., 2024). This paper focuses on the NeRF architectures used in Zama Ramirez et al. (2024) and Cardace et al. (2024), consisting of a single MLP and a tri-plane followed by an MLP, respectively, and the one proposed by Müller et al. (2022), i.e. a multi-resolution hash table followed by an MLP. In all three cases, the MLP adheres to the simplified formulation by Mildenhall et al. (2020) with no viewing direction, i.e. $(x, y, z) \mapsto (r, g, b, \sigma)$.

**Meta-networks.** Due to the high dimensionality of the weight space, its symmetries (Hecht-Nielsen, 1990), and the impact of randomness on the solution where training converges (Entezari et al., 2022; Ainsworth et al., 2023), processing neural network weights presents unique challenges that set them apart from more common input formats. The first works to address the design of neural networks that ingest the weights of other neural networks leverage group theory to devise architectures that are equivariant to the permutation symmetries of the input networks (Navon et al., 2023; Zhou et al., 2023a;b). Yet, these meta-networks are tailored to specific input networks, such as MLPs and CNNs without normalization layers, and cannot generalize to arbitrary input architectures. To overcome this limitation, Graph Meta-Networks (GMNs) were introduced (Lim et al., 2024; Kofinas et al., 2024). Since GMNs are graph neural networks, they are, by design, equivariant to the node permutations of input graphs and can ingest any graph. Therefore, the challenge of processing neural network weights turns into the task of transforming the input network into a graph. In this paper, we use the GMN formulation by Lim et al. (2024) and devise our own conversion of hash tables into graphs.

**Meta-networks for NeRF processing.** As the meta-network literature is still in its infancy, none of the aforementioned works include NeRFs as input in their experimental evaluation and instead choose to focus on simpler neural networks. The first methods to perform tasks on NeRFs by ingesting their weights are `nf2vec` (Zama Ramirez et al., 2024) and the framework by Cardace et al. (2024). `nf2vec` is an encoder-decoder architecture trained end-to-end with a rendering loss; at inference time, the encoder takes the weights of a NeRF as input and produces an embedding, which in turn becomes the input to traditional deep learning pipelines for downstream tasks. More recent works (Ballerini et al., 2024; Amaduzzi et al., 2024) investigate the potential applications of this approach to language-related tasks. While `nf2vec` is designed to ingest MLPs, Cardace et al. processes tri-planar NeRFs (Chan et al., 2022) by discarding the MLP and giving the tri-planar component alone as input to a Transformer. Yet, both `nf2vec` and Cardace et al. suffer from the same drawback as the first meta-networks: they are designed to handle specific NeRF architectures. In this work, we instead present the first architecture-agnostic framework for NeRF processing.

**Contrastive learning.** Contrastive learning is a representation learning approach that trains models to distinguish between similar (positive) and dissimilar (negative) data pairs by aiming to embed similar data points closer together while pushing dissimilar ones farther apart in latent space (Chen et al., 2020; He et al., 2020). Multimodal vision-language models extend this concept to align image and text modalities by maximizing the similarity between matching image-text pairs and minimizing it for mismatched ones, as demonstrated in CLIP (Radford et al., 2021), a model that learns a shared embedding space supporting zero-shot transfer to diverse vision tasks. Zhai et al. (2023) build upon this foundation and propose to replace the softmax-based loss used in CLIP with a simple pairwise sigmoid loss, called SigLIP, which is shown to work better for relatively small (4k–32k) batch sizes. For this reason, in this paper, we use the SigLIP loss to align GMN embeddings of different NeRF architectures representing the same object.

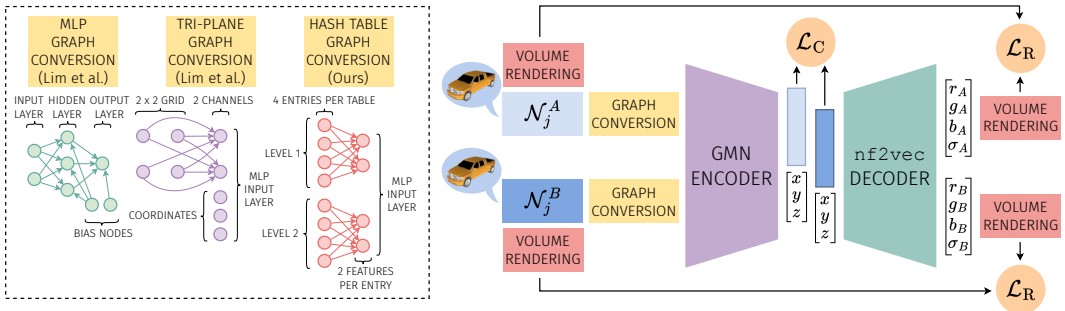

Figure 2: **Method overview. Left:** parameter graph construction for an MLP (left), a tri-plane (middle), and a multi-resolution hash table (right). For better clarity, the graphs of a single $2 \times 2 \times 2$ plane and of two $4 \times 2$ hash tables are shown. **Right:** our framework leverages a Graph Meta-Network (Lim et al., 2024) encoder alongside the `nf2vec` decoder (Zama Ramirez et al., 2024) and is trained end-to-end on a dataset of NeRFs with different architectures ($\mathcal{N}_j^A, \mathcal{N}_j^B$) with both a rendering ($\mathcal{L}_R$) and a contrastive ($\mathcal{L}_C$) loss.

## 3 METHOD

We tackle the challenging problem of embedding NeRFs parameterized by different neural architectures with a representation learning framework. Our method uses an encoder and a decoder trained end-to-end with a combination of rendering and contrastive objectives, where the encoder is implemented as a Graph Meta-Network (GMN). After training, the frozen encoder converts NeRF weights into an embedding that can serve as input to deep learning pipelines for downstream tasks. In the remainder of this section, we will describe each framework component; further details are provided in App. B.

**From NeRFs to graphs.** In order for a NeRF to be ingested by the encoder, it must be converted into a graph. The naive approach to perform this conversion would be to adopt the standard computation graph formulation, namely representing a neural network as a Directed Acyclic Graph (DAG), where nodes are activations and edges hold weight values. However, computation graphs scale poorly with the number of activations in networks with weight-sharing schemes, as a single weight requires multiple edges, one for each activation it affects. This limitation has motivated Lim et al. (2024) to introduce the *parameter graph* representation, where each weight is associated with a single edge of the graph, rather than multiple edges. Lim et al. describe the parameter graph construction of several common neural layers, which can then be concatenated to form the overall parameter graph. Among these layers, they provide the parameter graph conversion of spatial parameter grids, such as those often used in NeRF architectures, specifically tri-planar ones (Chan et al., 2022). However, they do not extend their parameter graph formulation to multi-resolution hash tables (Müller et al., 2022). In this paper, we leverage Lim et al.'s graph representations of linear layers and tri-planes, detailed in App. A, and propose a conversion into parameter graphs for hash tables, described here. These three graph representations are shown in Fig. 2 (left). Specifically, hash tables map spatial locations of a dense 3D grid to feature vectors stored in a table via a hashing function; in practice, multiple separate hash tables are used to index grids at different resolutions (hence the *multi-resolution* terminology). To compute the output, the feature vectors of the grid vertices enclosing a query point $(x, y, z)$ are interpolated at each resolution, and the results are concatenated to serve as input to an MLP. The most direct way to convert a hash table into a parameter graph would be to explicitly model the 3D grids of features encoded by the tables and adopt a graph representation analogous to the one used for tri-planes. Albeit simple, explicitly modeling the underlying voxel grid would require as many nodes as there are spatial locations, i.e. it would scale cubically with the resolution. Instead, we construct the parameter graph of a hash table with a node for each table entry and a node for each feature vector dimension. Then, we connect each entry node to each feature node via an edge that stores the corresponding feature value. This subgraph construction is repeated for each hash table, i.e. for each resolution level. Finally, the concatenation of all feature nodes becomes the first layer of the MLP parameter graph that follows. This graph representation preserves the memory efficiency of the hash table and is the one we adopt.

**Encoder.** Our framework's encoder is the GMN proposed by Lim et al. (2024), i.e. a standard message-passing Graph Neural Network (GNN) (Battaglia et al., 2018) with node and edge features but no global graph-level feature. Node features are updated by message-passing along neighbors and contribute to updating the features of the edges connecting them. The final embedding is obtained via an average pooling of the edge features. Notably, as the encoder is a GNN, it can process any input graph and, hence, any neural network that has undergone the previously described parameter-graph conversion, thereby allowing our framework to handle any NeRF architecture for which a graph representation is known.

**Decoder.** Our framework leverages the decoder first introduced by nf2vec (Zama Ramirez et al., 2024), which takes as input the concatenation of the embedding produced by the encoder alongside a frequency encoding (Mildenhall et al., 2020) of a 3D point $(x, y, z)$ and outputs a learned approximation of the radiance field value $(r, g, b, \sigma)$ at that point. Thus, the combination of decoder and embedding can itself be seen as a conditioned neural radiance field. Inspired by Park et al. (2019), the decoder architecture consists of a simple succession of linear layers, intertwined with ReLU activations, and a single skip connection from the input to halfway through the network.

**Training.** The encoder and the decoder are trained end-to-end with a combination of two loss terms: a rendering loss and a contrastive loss. The rendering loss is the one used by Zama Ramirez et al. (2024) to train the nf2vec framework and can be described as follows. Consider a NeRF $\mathcal{N} \colon \mathbf{x} \mapsto (\mathbf{c}, \sigma)$ with $\mathbf{x} = (x, y, z)$ and $\mathbf{c} = (r, g, b)$ and let $I \in \mathcal{I}$ be one of the images $\mathcal{N}$ was trained on, with corresponding camera pose and intrinsic parameters. Let $\mathcal{X} = \{\mathbf{x}_i\}$ be the set of points sampled along a ray cast from image $I$ into the scene and passing through pixel $\mathbf{p}$ on the image plane, and let $C_{\mathcal{N}}(\mathbf{p})$ be the color computed via volume rendering by accumulating the contributions of the $(\mathbf{c}_i, \sigma_i)$ output values of $\mathcal{N}$ for all inputs $\mathbf{x}_i \in \mathcal{X}$. Analogously, let $C_{\mathrm{D}}(\mathbf{p})$ be the color computed with the output values produced by the decoder when the encoder takes $\mathcal{N}$ as input. The rendering loss associated with NeRF $\mathcal{N}$ is defined as

$$\mathcal{L}_{\mathrm{R}}(\mathcal{N}) = \sum_{I \in \mathcal{I}} \sum_{\mathbf{p} \in \mathcal{S}(I)} \mathrm{smoothL1}(C_{\mathcal{N}}(\mathbf{p}), C_{\mathrm{D}}(\mathbf{p})) \tag{1}$$

where $\mathcal{S}(\mathcal{I})$ is a subset of pixels sampled from $\mathcal{I}$ and $\mathrm{smoothL1}$ is the smooth L1 loss (Girshick, 2015). More precisely, Zama Ramirez et al. compute one rendering loss term for foreground and one for background pixels, and express $\mathcal{L}_{\mathrm{R}}$ as a weighted sum of the two. Let $\mathcal{M}, \mathcal{T}, \mathcal{H}$ be three NeRF architectures and consider a dataset of NeRFs of 3D objects where each object $j$ appears three times: as a NeRF $\mathcal{M}_j$ parameterized by $\mathcal{M}$, as a NeRF $\mathcal{T}_j$ parameterized by $\mathcal{T}$, and as a NeRF $\mathcal{H}_j$ parameterized by $\mathcal{H}$. Given a mini-batch $\mathcal{B} = \{(\mathcal{N}_1^A, \mathcal{N}_1^B), (\mathcal{N}_2^A, \mathcal{N}_2^B), \dots\}$, where each $(\mathcal{N}_j^A, \mathcal{N}_j^B)$ is randomly sampled from $\{(\mathcal{M}_j, \mathcal{T}_j), (\mathcal{T}_j, \mathcal{H}_j), (\mathcal{M}_j, \mathcal{H}_j)\}$, the rendering loss computed over $\mathcal{B}$ is the average rendering loss of each NeRF $\mathcal{N}$ in $\mathcal{B}$, i.e.

$$\mathcal{L}_{\mathrm{R}} = \frac{1}{2|\mathcal{B}|} \sum_{j=1}^{|\mathcal{B}|} \sum_{\mathcal{N} \in (\mathcal{N}_j^A, \mathcal{N}_j^B)} \mathcal{L}_{\mathrm{R}}(\mathcal{N}) \tag{2}$$

where $\mathcal{L}_{\mathrm{R}}(\mathcal{N})$ is defined in Eq. 1. Our contrastive loss follows instead the definition by Zhai et al. (2023), namely

$$\mathcal{L}_{\mathrm{C}} = -\frac{1}{|\mathcal{B}|} \sum_{j=1}^{|\mathcal{B}|} \sum_{k=1}^{|\mathcal{B}|} \ln \frac{1}{1 + e^{-\ell_{jk}(t \mathbf{u}_j \cdot \mathbf{v}_k + b)}} \tag{3}$$

where $\ell_{jk} = 1$ if $j = k$ and $-1$ otherwise, $t$ and $b$ are learnable scalar hyperparameters, and $\mathbf{u}_j$ and $\mathbf{v}_k$ are the L2-normalized encoder embeddings of $\mathcal{N}_j^A$ and $\mathcal{N}_k^B$, respectively. Finally, the combined loss used to train our framework is

$$\mathcal{L}_{\mathrm{R+C}} = \mathcal{L}_{\mathrm{R}} + \lambda \mathcal{L}_{\mathrm{C}} \tag{4}$$

where $\lambda$ is a fixed hyperparameter. The rationale behind the choice of this loss is discussed in Sec. 4.1. Fig. 2 (right) shows an overview of our training procedure.

**Inference.** At inference time, a single forward pass of the encoder converts the parameter graph of a NeRF into a latent vector, which we then use as input to deep learning pipelines for classification, retrieval, and language tasks, as outlined in Fig. 1 and detailed in Secs. 4.2 to 4.4.

Table 1: **Dataset architectures.** Architectural hyperparameters of NeRF datasets featured in our experiments, including those used at inference time only (*unseen*).

| | Training | | | Unseen | | | | | | | | | |
| | MLP | TRI | HASH | MLP-2L | MLP-32H | TRI-2L | TRI-32H | TRI-16W | TRI-8C | HASH-2L | HASH-32H | HASH-3N | HASH-11T |
|---|---|---|---|---|---|---|---|---|---|---|---|---|---|
| MLP Hidden Layers | 3 | 3 | 3 | 2 | 3 | 2 | 3 | 3 | 3 | 2 | 3 | 3 | 3 |
| MLP Hidden Dim | 64 | 64 | 64 | 64 | 32 | 64 | 32 | 64 | 64 | 64 | 32 | 64 | 64 |
| Tri-plane Resolution | – | 32 | – | – | – | 32 | 32 | 16 | 32 | – | – | – | – |
| Tri-plane Channels | – | 16 | – | – | – | 16 | 16 | 16 | 8 | – | – | – | – |
| Hash Table Levels | – | – | 4 | – | – | – | – | – | – | 4 | 4 | 3 | 4 |
| Hash Table Size ($\log_2$) | – | – | 12 | – | – | – | – | – | – | 12 | 12 | 12 | 11 |

$$\mathcal{L}_{\mathrm{R}} \qquad\qquad \mathcal{L}_{\mathrm{R+C}} \qquad\qquad \mathcal{L}_{\mathrm{C}}$$

Figure 3: **t-SNE plots.** 2D projections of the latent space created by our framework when trained on a dataset of NeRFs of ShapenetRender objects (Xu et al., 2019), where each object is represented by three NeRFs parameterized by different architectures: MLPs, tri-planes, and multi-resolution hash tables.

## 4 EXPERIMENTS

**Datasets.** Our experimental evaluation is based on three families of NeRF architectures: MLP-based (i.e. a vanilla MLP), tri-planar (i.e. a tri-plane (Chan et al., 2022) followed by an MLP), and hash-based (i.e. a multi-resolution hash table (Müller et al., 2022) followed by an MLP). These families are exemplified by the following datasets: (i) one consisting of Zama Ramirez et al. (2024)'s MLP-based NeRFs only, which will be referred to as MLP; (ii) one consisting of Cardace et al. (2024)'s tri-planar NeRFs only, which will be referred to as TRI; (iii) one, created by us, consisting of hash-based NeRFs only, which will be referred to as HASH; (iv) the union of MLP, TRI, and HASH, i.e. a dataset where each object appears three times: once as an MLP-based NeRF, then as a tri-planar NeRF, and finally as a hash-based NeRF; we will refer to this dataset as ALL. To further assess the ability of our framework to perform tasks on arbitrary NeRF architectures, we create additional test sets of NeRFs featuring architectures belonging to the same families of those seen at training time but with different hyperparameters, which we will refer to as *unseen architectures*. The specifics of the architectures of all our datasets are reported in Tab. 1. Overall, we consider a total of 13 architectures across the three families. Following the protocol first introduced by Zama Ramirez et al. (2024), all the aforementioned NeRFs are trained on ShapenetRender (Xu et al., 2019), a dataset providing RGB images of synthetic 3D objects together with their class label.

**Models.** To assess the impact of different losses on training, we introduce a distinction between three versions of our framework, depending on the learning objective: (i) $\mathcal{L}_{\mathrm{R}}$, where the framework has been trained with the rendering loss of Eq. 2 alone; (ii) $\mathcal{L}_{\mathrm{R+C}}$, where the framework has been trained with a combination of rendering and contrastive losses as in Eq. 4, with $\lambda = 2 \times 10^{-2}$ (as it leads to similar magnitudes in the two terms); (iii) $\mathcal{L}_{\mathrm{C}}$, where the framework has been trained with the contrastive loss of Eq. 3 alone.

**Single vs multi-architecture.** Our method is the first neural processing framework able to handle multiple NeRF architectures. Thus, the natural setting to test it is when it is trained on ALL; we will refer to this scenario as the *multi-architecture setting*. Yet, our method can also be used when the input NeRFs all share the same architecture, e.g. when our framework has been trained on MLP, TRI, or HASH only, by dropping the contrastive loss term. Therefore, to test its generality, we also perform experiments in such a scenario, which we will refer to as the *single-architecture setting*. In this setting, our approach can be compared to previous methods (Zama Ramirez et al., 2024; Cardace et al., 2024) that are limited to handling specific architectures.

Table 2: **NeRF classification (multi-architecture).** The encoder is trained on `ALL`; the classifier is trained on the datasets in column 2 and tested on those in columns 3–5.

| Method | Classifier Training Set | Accuracy (%) ↑ | | |
|---|---|---|---|---|
| | | MLP | TRI | HASH |
| nf2vec | – | – | – | – |
| Cardace et al. | | – | – | – |
| $\mathcal{L}_R$ (ours) | ALL | **93.6** | **94.0** | **92.4** |
| $\mathcal{L}_{R+C}$ (ours) | | 90.7 | 90.6 | 90.0 |
| $\mathcal{L}_R$ (ours) | MLP | **93.8** | 25.3 | 19.3 |
| $\mathcal{L}_{R+C}$ (ours) | | 91.5 | **58.6** | **56.6** |
| $\mathcal{L}_R$ (ours) | TRI | 11.4 | **93.8** | 9.3 |
| $\mathcal{L}_{R+C}$ (ours) | | **77.8** | 91.0 | **66.5** |
| $\mathcal{L}_R$ (ours) | HASH | 13.8 | **35.7** | **92.7** |
| $\mathcal{L}_{R+C}$ (ours) | | **54.1** | 35.5 | 90.6 |

Table 3: **NeRF classification (single-architecture).** The classifier is trained and tested on the same dataset on which the encoder is trained.

| Method | Encoder Training Set | Accuracy (%) ↑ |
|---|---|---|
| nf2vec | MLP | 92.1 |
| Cardace et al. | | – |
| $\mathcal{L}_R$ (ours) | | **93.6** |
| nf2vec | TRI | – |
| Cardace et al. | | 93.1 |
| $\mathcal{L}_R$ (ours) | | **94.0** |
| nf2vec | HASH | – |
| Cardace et al. | | – |
| $\mathcal{L}_R$ (ours) | | **92.5** |

## 4.1 LATENT SPACE ANALYSIS

To study the impact of different learning objectives on the organization of the resulting NeRF latent space, we apply t-SNE dimensionality reduction (Van der Maaten & Hinton, 2008) to the embeddings computed by our framework on NeRFs belonging to the test set of the `ALL` dataset. Fig. 3 shows the resulting bi-dimensional plots. Some interesting patterns can be noted. When trained to minimize $\mathcal{L}_R$ alone (Fig. 3, left), the encoder creates an embedding space that is clustered by class, even though class labels were not used at training time. This behavior is a byproduct of the loss, which enforces NeRFs encoding similar shapes and colors to lie nearby in the latent space. Another outcome one may expect is that the same object represented by different NeRF architectures is projected into nearby embeddings; however, this turns out not to be the case. Instead, three distinct clusters emerge for each class, each corresponding to a NeRF architecture, and these clusters are further away from each other than clusters sharing the same underlying architecture. This outcome shows that $\mathcal{L}_R$ alone does not directly encourage the model to align NeRF embeddings regardless of the input architecture. Conversely, when trained to minimize $\mathcal{L}_C$ alone (Fig. 3, right), the encoder significantly reduces the distance between different architectures representing the same class, but also provides a considerably less distinguishable clusterization by class compared to $\mathcal{L}_R$ case (especially for some classes, like *chairs*, *sofas*, and *tables*). Finally, $\mathcal{L}_{R+C}$ (Fig. 3, middle) strikes a balance between these latent space properties. Compared to $\mathcal{L}_R$, one macro-cluster per class is present, although hash tables tend to form a separate sub-cluster (which is especially evident for *airplanes* and *watercrafts*). Compared to $\mathcal{L}_C$, classes are more separated (although not as much as with $\mathcal{L}_R$), but the sub-clusterization of hash tables is more apparent. Thus, we expect $\mathcal{L}_R$ to be the best choice for tasks where the separation between classes is highly relevant; $\mathcal{L}_{R+C}$ and $\mathcal{L}_C$, on the other hand, are likely to be the most effective for tasks where strong invariance to the NeRF architecture is required, with $\mathcal{L}_{R+C}$ being preferable, as it preserves more separation between classes. For these reasons, the remainder of this section shows experimental results obtained with $\mathcal{L}_R$ and $\mathcal{L}_{R+C}$; results for $\mathcal{L}_C$ are reported in App. C.

## 4.2 NERF CLASSIFICATION

Once our framework has been trained, NeRF classification is performed by learning a downstream classifier $\mathcal{C}$ to predict the labels of the embeddings produced by the encoder; an overview of this procedure is shown in Fig. 1. We report results for two cases: classification of NeRF architectures on which our framework was trained and classification of unseen architectures. In both cases, since the multi-architecture setting requires evaluating methods trained on `ALL`, neither one of the previous works, i.e. `nf2vec` (Zama Ramirez et al., 2024) and Cardace et al. (2024), can be applied to this scenario, as they can only be trained on `MLP` and `TRI`, respectively.

**Training architectures.** Tab. 2 shows NeRF classification results in the multi-architecture setting on architectures used to train our framework. When $\mathcal{C}$ has been trained on `ALL`, $\mathcal{L}_R$ performs the best: this result is consistent with the better separation between clusters corresponding to different classes provided by $\mathcal{L}_R$, as shown in Fig. 3 (left) and discussed in Sec. 4.1. This trend can also be observed, as expected, when $\mathcal{C}$ is trained on either `MLP`, `TRI`, or `HASH` and then tested on the same dataset, replicating the previous scenario with smaller datasets. On the other hand, the introduction of a contrastive objective in $\mathcal{L}_{R+C}$ is key to performance whenever $\mathcal{C}$ is trained on a single-architecture

Table 4: **NeRF classification of unseen architectures (multi-architecture).** The encoder is trained on `ALL`; the classifier is trained on the datasets in column 2 and tested on those in columns 3–12, containing NeRF architectures unseen at training time.

| Method | Classifier Training Set | Accuracy (%) ↑ | | | | | | | | | |
|---|---|---|---|---|---|---|---|---|---|---|---|
| | | MLP-2L | MLP-32H | TRI-2L | TRI-32H | TRI-16W | TRI-8C | HASH-2L | HASH-32H | HASH-3N | HASH-11T |
| nf2vec | | – | – | – | – | – | – | – | – | – | – |
| Cardace et al. | – | – | – | – | – | – | – | – | – | – | – |
| $\mathcal{L}_R$ (ours) | ALL | **91.3** | **87.4** | **93.2** | **88.3** | 24.8 | **69.4** | **91.9** | **91.2** | **88.3** | 24.5 |
| $\mathcal{L}_{R+C}$ (ours) | | 85.9 | 83.8 | 87.0 | 84.1 | **72.1** | 30.8 | 89.2 | 87.4 | 86.8 | **27.8** |
| $\mathcal{L}_R$ (ours) | MLP | **91.3** | **86.1** | 22.6 | 23.2 | 7.9 | **21.0** | 19.7 | 20.6 | 21.8 | 7.6 |
| $\mathcal{L}_{R+C}$ (ours) | | 86.6 | 81.3 | **63.3** | **43.4** | **59.2** | 13.1 | **51.5** | **50.8** | **55.0** | **24.8** |
| $\mathcal{L}_R$ (ours) | TRI | 11.9 | 10.6 | **92.1** | **84.8** | 32.1 | **43.6** | 9.2 | 8.8 | 15.0 | 5.2 |
| $\mathcal{L}_{R+C}$ (ours) | | **60.7** | **58.0** | 86.9 | 83.2 | **63.9** | 30.2 | **59.8** | **61.5** | **66.3** | **28.5** |
| $\mathcal{L}_R$ (ours) | HASH | 10.7 | 7.4 | 33.9 | **36.3** | 19.1 | **23.0** | **91.6** | **91.4** | **87.8** | **29.2** |
| $\mathcal{L}_{R+C}$ (ours) | | **47.0** | **40.4** | **40.6** | 34.0 | **43.4** | 19.3 | 89.5 | 87.9 | 86.4 | 25.7 |

Table 5: **NeRF classification of unseen architectures (single-architecture).** The classifier is trained on the same dataset as the encoder and tested on the datasets in columns 3–12, containing NeRF architectures unseen at training time.

| Method | Encoder Training Set | Accuracy (%) ↑ | | | | | | | | | |
|---|---|---|---|---|---|---|---|---|---|---|---|
| | | MLP-2L | MLP-32H | TRI-2L | TRI-32H | TRI-16W | TRI-8C | HASH-2L | HASH-32H | HASH-3N | HASH-11T |
| nf2vec | | 63.7 | – | – | – | – | – | – | – | – | – |
| Cardace et al. | MLP | – | – | – | – | – | – | – | – | – | – |
| $\mathcal{L}_R$ (ours) | | **91.8** | **83.7** | 10.9 | 6.1 | 7.0 | 4.9 | 5.7 | 5.6 | 4.0 | 4.9 |
| nf2vec | | – | – | – | – | – | – | – | – | – | – |
| Cardace et al. | TRI | – | – | 93.5 | 92.0 | – | 68.6 | – | – | – | – |
| $\mathcal{L}_R$ (ours) | | 10.1 | 10.0 | 92.6 | 82.5 | 22.9 | 72.8 | 17.4 | 13.6 | 10.8 | 12.1 |
| nf2vec | | – | – | – | – | – | – | – | – | – | – |
| Cardace et al. | HASH | – | – | – | – | – | – | – | – | – | – |
| $\mathcal{L}_R$ (ours) | | 8.3 | 8.9 | 15.4 | 21.3 | 21.3 | 21.3 | 90.6 | 91.0 | 88.2 | 28.8 |

dataset (e.g. `MLP`) and tested on a dataset with a different architecture (e.g. `TRI` or `HASH`), as clusters of NeRFs belonging to the same class but parameterized by different architectures are closer in the embedding space than with $\mathcal{L}_R$ alone. Overall, our method consistently achieves remarkable accuracies when tested on architectures included in the training set of $\mathcal{C}$ (i.e. red cells in Tab. 2). In the single-architecture setting, instead, previous methods can be trained and evaluated on the datasets corresponding to the NeRF architecture they were designed to process, whereas $\mathcal{L}_{R+C}$ cannot be applied, as there is a single architecture and, therefore, no positive pairs to compute the contrastive loss. As shown in Tab. 3, when both the encoder and the classifier are trained and tested on either `MLP` or `TRI`, our method outperforms both `nf2vec` and Cardace et al., while also achieving high accuracy on `HASH`. Hence, beyond its original aim, our approach can also provide a competitive alternative to single-architecture frameworks.

**Unseen architectures.** Multi-architecture classification results on unseen architectures are shown in Tab. 4. Remarkably, a trend coherent with that of Tab. 2 can be noticed, namely that $\mathcal{L}_R$ tends to perform best when the classifier $\mathcal{C}$ is trained either on `ALL` or on an architecture belonging to the family of the unseen one presented at test time (i.e. orange cells in Tab. 4), whereas $\mathcal{L}_{R+C}$ almost always prevails when training and test families differ, usually by a significant margin. Notable exceptions are `TRI-16W`, where $\mathcal{L}_{R+C}$ consistently yields the best results, and `TRI-8C`, where $\mathcal{L}_R$ outperforms $\mathcal{L}_{R+C}$ (although less dramatically) even when changing the architecture family. Some other interesting patterns can be noted: in MLP and tri-plane families, reducing the number of hidden layers of an MLP has less impact on accuracy than reducing the hidden dimension, whereas it does not seem to have a significant effect on hash table families; furthermore, changing hash table size is the most disruptive change, even when encoder and classifier have seen hash-based NeRFs at training time. These patterns can serve as the basis for future investigations into this topic. Overall, it is worth highlighting that, in the most relevant scenario, i.e. when our framework has been trained on `ALL`, our method is able to reach remarkable accuracies on most of the unseen variations of known architecture families it is tested on. In the single-architecture setting, shown in Tab. 5, $\mathcal{L}_R$ performs much better than `nf2vec` in the only case in which the latter can be tested. Conversely, $\mathcal{L}_R$ performs worse than Cardace et al. (2024) in `TRI-2L` and `TRI-32H`, i.e. `TRI` variations that change the MLP architecture. This is not surprising since, when performing classification, Cardace et al. processes the tri-plane alone and discards the MLP, which makes their approach invariant to changes in the MLP architecture. Yet, since their method is tied to the tri-plane quantization, it cannot process `TRI-16W` and is less robust than ours to the reduction in the number of channels. Finally, our method is the only one that can be tested on the hash table family, where it exhibits robust performance when trained on

Table 6: **NeRF retrieval (ShapenetRender).** The encoder is trained on `ALL`. Query/gallery combinations belong to their respective test sets.

| Method | k | MLP/TRI | MLP/HASH | TRI/MLP | TRI/HASH | HASH/MLP | HASH/TRI |
|---|---|---|---|---|---|---|---|
| nf2vec (Cardace et al.) | – | – | – | – | – | – | – |
| RANDOM | 1 | 0.03 | 0.03 | 0.03 | 0.03 | 0.03 | 0.03 |
| $\mathcal{L}_R$ (ours) | | 1.80 | 0.43 | 4.48 | 1.67 | 1.06 | 0.38 |
| $\mathcal{L}_{R+C}$ (ours) | | **30.62** | **14.77** | **33.27** | **13.17** | **13.45** | **9.46** |
| RANDOM | 5 | 0.13 | 0.13 | 0.13 | 0.13 | 0.13 | 0.13 |
| $\mathcal{L}_R$ (ours) | | 5.28 | 1.82 | 13.30 | 6.12 | 3.84 | 1.37 |
| $\mathcal{L}_{R+C}$ (ours) | | **59.37** | **37.98** | **61.24** | **33.40** | **32.09** | **25.13** |
| RANDOM | 10 | 0.25 | 0.25 | 0.25 | 0.25 | 0.25 | 0.25 |
| $\mathcal{L}_R$ (ours) | | 8.19 | 3.08 | 19.24 | 9.23 | 6.09 | 2.68 |
| $\mathcal{L}_{R+C}$ (ours) | | **72.67** | **51.25** | **72.62** | **45.01** | **42.83** | **36.31** |

Table 7: **NeRF retrieval (Objaverse generalization).** The encoder is trained on `ALL`. Query/gallery combinations belong to their respective test sets.

| Method | k | MLP$^{OB}$/TRI$^{OB}$ | MLP$^{OB}$/HASH$^{OB}$ | TRI$^{OB}$/MLP$^{OB}$ | TRI$^{OB}$/HASH$^{OB}$ | HASH$^{OB}$/MLP$^{OB}$ | HASH$^{OB}$/TRI$^{OB}$ |
|---|---|---|---|---|---|---|---|
| nf2vec (Cardace et al.) | – | – | – | – | – | – | – |
| RANDOM | 1 | 0.05 | 0.05 | 0.05 | 0.05 | 0.05 | 0.05 |
| $\mathcal{L}_R$ (ours) | | 2.60 | 0.48 | 2.71 | 0.48 | 0.74 | 0.32 |
| $\mathcal{L}_{R+C}$ (ours) | | **9.82** | **6.10** | **8.33** | **1.75** | **6.74** | **2.92** |
| RANDOM | 5 | 0.27 | 0.27 | 0.27 | 0.27 | 0.27 | 0.27 |
| $\mathcal{L}_R$ (ours) | | 6.37 | 1.11 | 7.27 | 1.91 | 2.92 | 0.69 |
| $\mathcal{L}_{R+C}$ (ours) | | **23.67** | **17.94** | **20.70** | **5.73** | **19.59** | **8.49** |
| RANDOM | 10 | 0.53 | 0.53 | 0.53 | 0.53 | 0.53 | 0.53 |
| $\mathcal{L}_R$ (ours) | | 9.66 | 2.39 | 10.99 | 3.24 | 4.88 | 1.80 |
| $\mathcal{L}_{R+C}$ (ours) | | **33.65** | **26.38** | **27.55** | **9.50** | **28.13** | **12.74** |

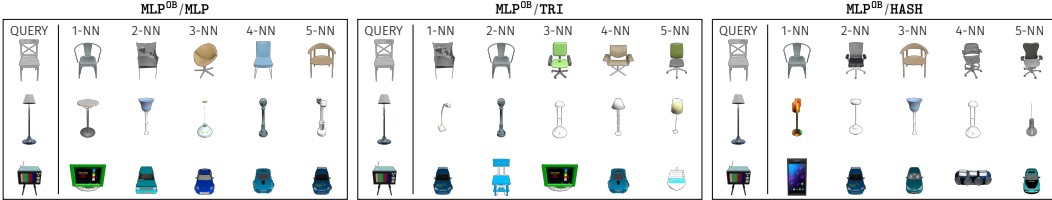

Figure 4: **Across-datasets NeRF retrieval** ($\mathcal{L}_{R+C}$). Query from the test set of `MLP`$^{OB}$ (Amaduzzi et al., 2025), gallery from the test set of `MLP`, `TRI`, or `HASH`.

`HASH`, except for `HASH-11T`, which already emerged as particularly challenging in the results of Tab. 4.

## 4.3 NERF RETRIEVAL

The embeddings produced by the trained encoder can also be used to perform retrieval tasks via $k$-nearest neighbor search, as outlined in Fig. 1. In particular, we define *instance-level* retrieval as follows: given a NeRF embedding of a given object (a.k.a. the *query*) and a *gallery* of NeRF embeddings, the goal is to find the embedding in the gallery that represents the same object as the query but encodes a different NeRF architecture. We evaluate the performance on this task with the recall@$k$ metric (Wang et al., 2017), defined as the percentage of queries whose $k$ nearest neighbors contain the NeRF representing the same object as the query. Tab. 6 shows the retrieval results for various query/gallery test-set combinations; the recall@$k$ obtained by randomly shuffling the gallery is shown for reference (RANDOM rows). Although both $\mathcal{L}_R$ and $\mathcal{L}_{R+C}$ significantly outperform the RANDOM baseline, the superior performance of $\mathcal{L}_{R+C}$ confirms that the contrastive objective favors a more architecture-agnostic latent space. Indeed, Figs. 6 and 7 display qualitatively that the latent space organization learned by $\mathcal{L}_{R+C}$ captures the NeRF similarity in color and shape independently of the underlying architecture.

**Generalization on Objaverse.** To assess the generalization capabilities of our framework, we leverage three additional datasets: `MLP`$^{OB}$, `TRI`$^{OB}$, and `HASH`$^{OB}$. With `MLP`$^{OB}$ we denote a dataset of MLP-based NeRFs trained on Objaverse (Deitke et al., 2023) recently released by Amaduzzi et al. (2025), whereas `TRI`$^{OB}$ and `HASH`$^{OB}$ are datasets, created by us, of tri-planar and hash-based NeRFs trained on the test split of Objaverse defined by Amaduzzi et al. Specifically, NeRFs in `MLP`$^{OB}$, `TRI`$^{OB}$, and `HASH`$^{OB}$ have the same architecture as those in `MLP`, `TRI`, and `HASH`, respectively (see Tab. 1). Tab. 7 mirrors the experiment in Tab. 6, with the notable difference that the query and gallery contain NeRFs trained on a different dataset (i.e. Objaverse) than the one the model has seen during training (i.e. ShapenetRender (Xu et al., 2019)). Although this leads, unsurprisingly, to an overall drop in performance compared to Tab. 6, it is worth noting, once again, that both $\mathcal{L}_R$ and $\mathcal{L}_{R+C}$ outperform the RANDOM baseline, and the contrastive loss yields superior results. Fig. 4 shows instead a qualitative retrieval experiment where the query is an MLP-based NeRF trained on Objaverse (i.e. `MLP`$^{OB}$) while the gallery contains NeRFs trained on ShapenetRender, hence the query object is not present in the gallery. Remarkably, for queries that resemble objects featured in ShapenetRender, our method retrieves NeRFs that belong to the same class as the query (rows 1–2 of Fig. 4), with some failure cases occurring with more unusual queries (row 3). An extended version of these results is shown in Fig. 8.

Table 8: **NeRF brief captioning (multi-architecture).**

| Method | Test Dataset | S-BERT ↑ | SimCSE ↑ |
|---|---|---|---|
| LLaNA | – | – | – |
| $\mathcal{L}_{\text{R+C}}$ (ours) | MLP | **67.1** | **68.9** |
| | TRI | 66.4 | 67.8 |
| | HASH | 66.3 | 67.9 |

Table 9: **NeRF detailed captioning (multi-architecture).**

| Method | Test Dataset | S-BERT ↑ | SimCSE ↑ |
|---|---|---|---|
| LLaNA | – | – | – |
| $\mathcal{L}_{\text{R+C}}$ (ours) | MLP | **73.3** | **75.1** |
| | TRI | 72.2 | 73.8 |
| | HASH | 72.4 | 74.0 |

Table 10: **NeRF single-round Q&A (multi-architecture).**

| Method | Test Dataset | S-BERT ↑ | SimCSE ↑ |
|---|---|---|---|
| LLaNA | – | – | – |
| $\mathcal{L}_{\text{R+C}}$ (ours) | MLP | **80.9** | **81.4** |
| | TRI | 80.7 | 81.3 |
| | HASH | 80.6 | 81.3 |

Table 11: **NeRF brief captioning (single-architecture).**

| Method | S-BERT ↑ | SimCSE ↑ |
|---|---|---|
| LLaNA | **68.6** | **70.5** |
| $\mathcal{L}_{\text{R}}$ (ours) | 68.3 | 70.2 |

Table 12: **NeRF detailed captioning (single-architecture).**

| Method | S-BERT ↑ | SimCSE ↑ |
|---|---|---|
| LLaNA | **77.4** | **79.8** |
| $\mathcal{L}_{\text{R}}$ (ours) | 74.1 | 76.0 |

Table 13: **NeRF single-round Q&A (single-architecture).**

| Method | S-BERT ↑ | SimCSE ↑ |
|---|---|---|
| LLaNA | **81.0** | **81.6** |
| $\mathcal{L}_{\text{R}}$ (ours) | **81.0** | **81.6** |

## 4.4 NeRF Captioning and Q&A

Amaduzzi et al. (2024; 2025) have previously shown that language tasks on NeRFs benefit from the holistic understanding of 3D objects captured by NeRF weights, leading their weight-processing framework (dubbed LLaNA) to outperform methods ingesting rendered views as input in NeRF captioning and NeRF Q&A tasks. To achieve these results, LLaNA uses the embeddings produced by the trained nf2vec encoder to condition an LLM belonging to the Llama 2 family (Touvron et al., 2023). To assess the effectiveness of our learned embeddings in downstream language tasks, we replaced the nf2vec encoder in LLaNA's pipeline with our GMN encoder, trained the resulting framework on textual descriptions from the ShapeNeRF–Text dataset (Amaduzzi et al., 2024), and evaluated it on brief captioning, detailed captioning, and single-round Q&A. In the multi-architecture setting, our framework is trained on ALL (as usual), and the ALL embeddings produced by our encoder are used to train the LLM. Since the nf2vec encoder can process only MLP-based NeRFs, the original LLaNA framework cannot be applied in the multi-architecture setting. In the single-architecture setting, instead, our framework is trained on MLP, and the MLP embeddings produced by our encoder are used to train the LLM. In this setting, our results are compared with the original LLaNA framework, which is also trained on MLP. Multi-architecture results are shown in Tabs. 8 to 10, whereas single-architecture results are shown in Tabs. 11 to 13. As in Amaduzzi et al. (2024; 2025), the reported metrics are a measure of the similarity between the global embeddings of the generated and ground-truth texts, where such embeddings are produced by pre-trained Sentence-BERT (Reimers & Gurevych, 2019) and SimCSE (Gao et al., 2021) encoders, respectively. Extended results including traditional n-gram-based metrics are shown in App. F. Two main conclusions can be drawn from Tabs. 8 to 13: in the multi-architecture setting, our method is remarkably robust to the choice of test-time NeRF architecture; when instead restricted to MLP-based NeRFs only, our method achieves results comparable to those of the original LLaNA framework. These results further confirm the architecture-agnostic nature of our embeddings and their effectiveness in downstream tasks, even as complex as those involving multimodal NeRF–language understanding.

## 5 Conclusions

In this paper, we presented the first framework that performs downstream tasks on NeRF weights with diverse architectures. Our method, based on a Graph Meta-Network encoder ingesting the NeRF parameter graph, can (i) handle input NeRFs parameterized by MLPs, tri-planes, and, for the first time, multi-resolution hash tables; (ii) process, at inference time, variations of architectures seen during training; (iii) provide state-of-the-art performance in the single-architecture scenario. Furthermore, we investigated the interplay between rendering and contrastive training objectives and showed that they serve complementary purposes, by favoring either class-level separability or invariance to the NeRF architecture. The main limitation of our study lies in its experimental evaluation, which primarily focuses on NeRFs that, although parameterized by 13 total architectures, are trained on a single dataset, i.e. ShapenetRender (Xu et al., 2019), with the notable exception of two generalization experiments on Objaverse (Deitke et al., 2023). Extending our training procedure on Objaverse itself or similar large-scale NeRF datasets would further validate the effectiveness of our approach, and we plan to follow this research path in future work. Eventually, our methodology could be scaled up to become the first foundational model for NeRF weight space processing.

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

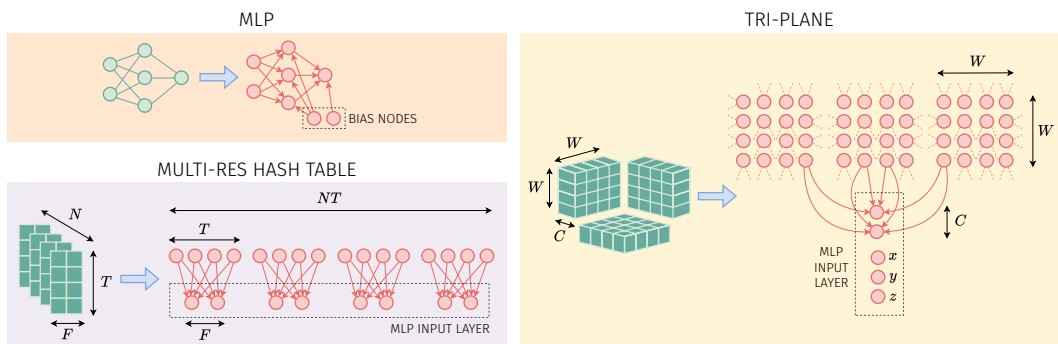

Figure 5: **Parameter graph conversion. Top left:** parameter graph representation of an MLP, proposed by Lim et al. (2024). **Right:** parameter graph representation of a tri-plane, proposed by Lim et al. (2024). Dotted edges should be connected to the $C$ channel nodes, but are not fully drawn for better visual clarity. **Bottom left:** our parameter graph representation of a multi-resolution hash table.

## A  ADDITIONAL PARAMETER GRAPH DETAILS

The parameter graph conversion of an MLP, a tri-plane, and a multi-resolution hash table is depicted in Fig. 5, with additional details compared to Fig. 2 (left).

**MLP.** The parameter graph of a linear layer (proposed by Lim et al. (2024)) coincides with its computation graph, except for biases, which are modeled by including an additional node for each (non-input) layer, connected to every other neuron in the layer with an edge. As a result, the parameter graph of an MLP is identical to its computation graph, with the addition of the aforementioned bias nodes. Each node stores three integer values: layer index, neuron index within the layer, and node type (*neuron* or *bias*). Each edge stores the corresponding weight or bias and two integers, i.e. layer index and edge type (*linear layer weight* or *linear layer bias*).

**Tri-plane.** The parameter graph of a tri-plane (proposed by Lim et al. (2024)) contains one node for each spatial location and one node for each feature channel, where each spatial node is connected via an edge to every channel node; the channel nodes, together with one node for each input coordinate (more precisely, one node for each entry in their frequency encoding; see App. B) become the first layer of the MLP parameter graph that follows. Each node stores three integer values: layer index, neuron index within the layer, and node type (*neuron* for coordinate nodes or *tri-planar* for spatial and channel nodes). Each edge stores the corresponding learnable parameter (a.k.a. feature) and five additional values: layer index (integer), edge type (*tri-planar*, integer), and three $(i, j, k)$ values (evenly spaced floats $\in [-1, 1]$) denoting the spatial position of the parameter within the tri-plane (with $i = 0$ on the $yz$ plane, $j = 0$ in the $xz$ plane, and $k = 0$ on the $xy$ plane).

**Hash table.** As mentioned in Sec. 3, we are the first to propose a parameter graph construction for multi-resolution hash tables. A single hash table is modeled with a node for each table entry and a node for each feature vector dimension, where each entry node is connected to each feature node via an edge. This subgraph construction is repeated for each hash table, i.e. for each resolution level. Finally, the concatenation of all feature nodes becomes the first layer of the MLP parameter graph that follows. Each node stores three integer values: layer index, neuron index within the layer, and node type (*hash table*). Each edge stores the corresponding learnable parameter (a.k.a. feature) and four additional integers: layer index, edge type (*hash table*), table index, and entry index within the table. Consider a multi-resolution hash table with $N$ levels, max table size $T$, $F$ features per entry, and max resolution $R$. Our parameter graph construction requires $N(T + F)$ nodes and $NTF$ edges. Conversely, if we drew an analogy with the tri-planar graph representation and explicitly modeled the underlying voxel grids with one node for each spatial location, it would result in a graph with $\mathcal{O}(N(R^3 + F))$ nodes and $\mathcal{O}(NR^3F)$ edges. Since the purpose of hash grids is to reduce the memory footprint by mapping multiple voxel grid locations to the same table entry via a hashing function, $T$ should always be much smaller than $R^3$ (e.g. NeRFs in HASH have $T = 2^{12}$ vs $R^3 = 2^{21}$), hence $N(T + F) \ll N(R^3 + F)$ and $NTF \ll NR^3F$. Therefore, the motivation behind our choice of parameter graph representation for hash tables lies in its higher memory efficiency compared to its voxel-grid-based alternative.

Table 14: **Dataset architectures.** Extended version of Tab. 1. Architectural hyperparameters of NeRF datasets featured in our experiments, including those used at inference time only (*unseen*).

| | Training | | | Unseen (ours) | | | | | | | | | |
|---|---|---|---|---|---|---|---|---|---|---|---|---|---|
| | MLP | TRI | HASH | MLP-2L | MLP-32H | TRI-2L | TRI-32H | TRI-16W | TRI-8C | HASH-2L | HASH-32H | HASH-3N | HASH-11T |
| Frequency Encoding | ✓ | ✓ | ✗ | ✓ | ✓ | ✓ | ✓ | ✓ | ✓ | ✗ | ✗ | ✗ | ✗ |
| MLP Activation | ReLU | Sine | ReLU | ReLU | ReLU | Sine | Sine | Sine | Sine | ReLU | ReLU | ReLU | ReLU |
| MLP Hidden Layers | 3 | 3 | 3 | 2 | 3 | 2 | 3 | 3 | 3 | 2 | 3 | 3 | 3 |
| MLP Hidden Dimension | 64 | 64 | 64 | 64 | 32 | 64 | 32 | 64 | 64 | 64 | 32 | 64 | 64 |
| Tri-plane Resolution | – | 32 | – | – | – | 32 | 32 | 16 | 32 | – | – | – | – |
| Tri-plane Channels | – | 16 | – | – | – | 16 | 16 | 16 | 8 | – | – | – | – |
| Hash Table Levels | – | – | 4 | – | – | – | – | – | – | 4 | 4 | 3 | 4 |
| Hash Table Size ($\log_2$) | – | – | 12 | – | – | – | – | – | – | 12 | 12 | 12 | 11 |
| Hash Table Features per Entry | – | – | 2 | – | – | – | – | – | – | 2 | 2 | 2 | 2 |

# B  ADDITIONAL EXPERIMENTAL DETAILS

**NeRF datasets.** As outlined in Sec. 4, the datasets used throughout our experiments mostly consist of NeRFs trained on ShapenetRender (Xu et al., 2019). ShapenetRender is a dataset featuring RGB images and class labels of synthetic objects belonging to 13 classes of the ShapeNetCore dataset (Chang et al., 2015): *airplane*, *bench*, *cabinet*, *car*, *chair*, *display*, *lamp*, *speaker*, *rifle*, *sofa*, *table*, *phone*, and *watercraft*. For each object, the dataset contains 36 $224 \times 224$ images. We follow Zama Ramirez et al. (2024)'s train-val-test split without augmentations, consisting of 31744 NeRFs used for training, 3961 for validation, and 3961 for testing. Tab. 14 summarizes the NeRF architectures featured in our experiments. In addition to the dataset descriptions provided in Sec. 4, we note here that:

- In `MLP` and `TRI`, input $(x, y, z)$ coordinates go through a frequency encoding (Mildenhall et al., 2020) before being concatenated to the tri-plane or hash table output that gets fed to the MLP, as this choice was made by the original creators of the two datasets, i.e. Zama Ramirez et al. (2024) and Cardace et al. (2024), respectively. Conversely, the MLP in `HASH` takes as input the output of the multi-resolution hash table by itself, without (encoded) input coordinates being concatenated to it; we choose to do so in order to be faithful to the original multi-resolution hash table architecture proposed by Müller et al. (2022). The presence or absence of concatenated encoded coordinates in a dataset used for training is then reflected in all the unseen variations of that architecture family (e.g. `MLP-2L` uses frequency encoding, whereas `HASH-2L` does not).
- MLP layers are intertwined with ReLU activation functions in `MLP` and `HASH`, whereas `TRI` uses sine activation functions (Sitzmann et al., 2020). The activation choices for `MLP` and `TRI` were made by their original creators (Zama Ramirez et al., 2024; Cardace et al., 2024), whereas we chose ReLU activations for `HASH` to remain faithful to the original hash table architecture (Müller et al., 2022). The activation choice in a dataset used for training is then reflected in all the unseen variations of that architecture family (e.g. `MLP-2L` uses ReLU, whereas `TRI-2L` uses sine).
- All NeRFs approximate a function from input coordinates to predicted color and density values, i.e. $(x, y, z) \mapsto (r, g, b, \sigma)$. This follows the simplified NeRF formulation proposed by Mildenhall et al. (2020), the alternative being adding a viewing direction $(\theta, \phi)$ as input and letting the NeRF approximate a function $(x, y, z, \theta, \phi) \mapsto (r, g, b, \sigma)$. The formulation with viewing direction is usually modeled by two MLPs: a *density MLP* taking as input $(x, y, z)$ (or, more precisely, an encoding of $(x, y, z)$) and producing as output $\sigma$ together with an additional vector $\mathbf{h}$ of features, and a *color MLP* taking as input the concatenation of $\mathbf{h}$ and $(\theta, \phi)$ and predicting $(r, g, b)$ as output. Both Zama Ramirez et al. (2024) and Cardace et al. (2024) adopt the simplified $(x, y, z) \mapsto (r, g, b, \sigma)$ formulation modeled by a single MLP when creating their NeRF datasets (i.e. `MLP` and `TRI`, respectively). We choose to follow the same formulation for `HASH` and the datasets of unseen architectures, both to keep the datasets consistent and to reduce the complexity of the parameter graph going into the encoder.

Furthermore, the experiments in Sec. 4.3 leverage three additional test sets, dubbed `MLP`[OB], `TRI`[OB], and `HASH`[OB]. `MLP`[OB] is a dataset of MLP-based NeRFs trained on Objaverse (Deitke et al., 2023), a large-scale dataset of 3D objects. `MLP`[OB] was recently released by Amaduzzi et al. (2025), whereas we created `TRI`[OB] and `HASH`[OB] ourselves by training tri-planar and hash-based NeRFs on the Objaverse test split used in Amaduzzi et al. (2025). More precisely, our test split is the union of the two test splits featured in Amaduzzi et al. (2025)'s benchmark, i.e. the one adopted by PointLLM (Xu et al., 2024) and the one adopted by GPT4Point (Qi et al., 2024). The resulting test set contains 1884

Table 15: **NeRF classification (multi-architecture).** Extended version of Tab. 2 with added $\mathcal{L}_C$ results. The encoder is trained on `ALL`; the classifier is trained on the datasets in column 2 and tested on those in columns 3–5.

| Method | Classifier Training Set | Accuracy (%) ↑ | | |
|---|---|---|---|---|
| | | MLP | TRI | HASH |
| `nf2vec` (Zama Ramirez et al., 2024) | _ | – | – | – |
| Cardace et al. (2024) | | – | – | – |
| $\mathcal{L}_R$ (ours) | | **93.6** | **94.0** | **92.4** |
| $\mathcal{L}_{R+C}$ (ours) | `ALL` | 90.7 | 90.6 | 90.0 |
| $\mathcal{L}_C$ (ours) | | 82.5 | 77.8 | 64.0 |
| $\mathcal{L}_R$ (ours) | | **93.8** | 25.3 | 19.3 |
| $\mathcal{L}_{R+C}$ (ours) | `MLP` | 91.5 | 58.6 | **56.6** |
| $\mathcal{L}_C$ (ours) | | 83.2 | **60.9** | 38.6 |
| $\mathcal{L}_R$ (ours) | | 11.4 | **93.8** | 9.3 |
| $\mathcal{L}_{R+C}$ (ours) | `TRI` | **77.8** | 91.0 | **66.5** |
| $\mathcal{L}_C$ (ours) | | 70.4 | 79.0 | 47.6 |
| $\mathcal{L}_R$ (ours) | | 13.8 | 35.7 | **92.7** |
| $\mathcal{L}_{R+C}$ (ours) | `HASH` | 54.1 | 35.5 | 90.6 |
| $\mathcal{L}_C$ (ours) | | **65.4** | **63.7** | 64.6 |

**NeRFs.** NeRFs in $\text{MLP}^{OB}$, $\text{TRI}^{OB}$, and $\text{HASH}^{OB}$ have the same architecture as those in `MLP`, `TRI`, and `HASH`, respectively.

**Encoder architecture.** Our encoder is a slight re-adaptation of the graph meta-network used in Lim et al. (2024) in their experiment called *predicting accuracy for varying architectures*. It has the same network hyperparameters: hidden dimension 128, 4 layers, a pre-encoder, 2 readout layers, and uses directed edges. However, unlike Lim et al. (2024), before average-pooling the edge features, it maps them to vectors of size 1024 through a single linear layer, resulting in a final embedding of size 1024 as in `nf2vec` (Zama Ramirez et al., 2024).

**Decoder architecture.** We use the `nf2vec` decoder in its original formulation (Zama Ramirez et al., 2024). The input coordinates $(x, y, z)$ are mapped to a vector of size 144 through a frequency encoding (Mildenhall et al., 2020) and concatenated with the 1024-dimensional embedding produced by the encoder. The resulting vector of size 1168 becomes the input of the decoder, which consists of 4 hidden layers with dimension 1024 intertwined with ReLU activations, with a skip connection that maps the input vector to a vector of size 1024 via a single linear layer + ReLU and sums it to the output of the second hidden layer. Finally, the decoder outputs the four $(r, g, b, \sigma)$ predicted radiance field values.

**Training.** Our encoder-decoder framework is trained end-to-end for 250 epochs with batch size 8, AdamW optimizer (Loshchilov & Hutter, 2019), one-cycle learning rate scheduler (Smith & Topin, 2019), maximum learning rate $1 \times 10^{-4}$ and weight decay $1 \times 10^{-2}$. These training hyperparameters are the same for $\mathcal{L}_R$, $\mathcal{L}_{R+C}$, and $\mathcal{L}_C$. In Eq. 1, foreground and background pixels are weighted by 0.8 and 0.2, respectively, as done in Zama Ramirez et al. (2024). In Eq. 3, $t$ and $b$ are initialized to 10 and $-10$, respectively, as done in Zhai et al. (2023). In Eq. 4, $\lambda$ is set to $2 \times 10^{-2}$, as we experimentally observed that this choice leads to $\mathcal{L}_R$ and $\mathcal{L}_C$ values with the same order of magnitude.

**Classification.** In our classification experiments of Sec. 4.2, the classifier is a concatenation of 3 (linear → batch-norm → ReLU → dropout) blocks, where the hidden dimensions of the linear layers are 1024, 512, and 256, and a final linear layer at the end computes the class logits. The classifier is trained via cross-entropy loss for 150 epochs with batch size 256, AdamW optimizer (Loshchilov & Hutter, 2019), one-cycle learning rate scheduler (Smith & Topin, 2019), maximum learning rate $1 \times 10^{-4}$, and weight decay $1 \times 10^{-2}$. These same network and training hyperparameters are used in the classification experiments of Zama Ramirez et al. (2024).

## C  $\mathcal{L}_C$ CLASSIFICATION AND RETRIEVAL RESULTS

**NeRF classification.** Tabs. 15 and 16 show classification results in the multi-architecture setting, including those produced by $\mathcal{L}_C$, which are omitted in Tabs. 2 and 4. Specifically, Tab. 15 shows

Table 16: **NeRF classification of unseen architectures (multi-architecture).** Extended version of Tab. 4 with added $\mathcal{L}_C$ results. The encoder is trained on ALL; the classifier is trained on the datasets in column 2 and tested on those in columns 3–12, containing NeRF architectures unseen at training time.

| Method | Classifier Training Set | Accuracy (%) ↑ | | | | | | | | | |
|---|---|---|---|---|---|---|---|---|---|---|---|
| | | MLP-2L | MLP-32H | TRI-2L | TRI-32H | TRI-16W | TRI-8C | HASH-2L | HASH-32H | HASH-3N | HASH-11T |
| nf2vec (Zama Ramirez et al., 2024) | – | – | – | – | – | – | – | – | – | – | – |
| Cardace et al. (2024) | – | – | – | – | – | – | – | – | – | – | – |
| $\mathcal{L}_R$ (ours) | | **91.3** | **87.4** | **93.2** | **88.3** | 24.8 | 69.4 | **91.9** | **91.2** | **88.3** | 24.5 |
| $\mathcal{L}_{R+C}$ (ours) | ALL | 85.9 | 83.8 | 87.0 | 84.1 | **72.1** | 30.8 | 89.2 | 87.4 | 86.8 | **27.8** |
| $\mathcal{L}_C$ (ours) | | 67.3 | 47.7 | 63.5 | 43.3 | 60.4 | 4.6 | 51.9 | 43.5 | 55.9 | 27.7 |
| $\mathcal{L}_R$ (ours) | | **91.3** | **86.1** | 22.6 | 23.2 | 7.9 | **21.0** | 19.7 | 20.6 | 21.8 | 7.6 |
| $\mathcal{L}_{R+C}$ (ours) | MLP | 86.6 | 81.3 | **63.3** | **43.4** | **59.2** | 13.1 | **51.5** | **50.8** | **55.0** | 24.8 |
| $\mathcal{L}_C$ (ours) | | 69.6 | 52.6 | 59.7 | 42.0 | 55.2 | 6.0 | 33.0 | 23.3 | 39.4 | **29.5** |
| $\mathcal{L}_R$ (ours) | | 11.9 | 10.6 | **92.1** | **84.8** | 32.1 | **43.6** | 9.2 | 8.8 | 15.0 | 5.2 |
| $\mathcal{L}_{R+C}$ (ours) | TRI | **60.7** | **58.0** | 86.9 | 83.2 | **63.9** | 30.2 | **59.8** | **61.5** | **66.3** | **28.5** |
| $\mathcal{L}_C$ (ours) | | 54.6 | 37.5 | 62.8 | 44.9 | 57.3 | 5.8 | 36.2 | 27.9 | 43.1 | 28.4 |
| $\mathcal{L}_R$ (ours) | | 10.7 | 7.4 | 33.9 | **36.3** | 19.1 | **23.0** | **91.6** | **91.4** | **87.8** | **29.2** |
| $\mathcal{L}_{R+C}$ (ours) | HASH | 47.0 | **40.4** | 40.6 | 34.0 | **43.4** | 19.3 | 89.5 | 87.9 | 86.4 | 25.7 |
| $\mathcal{L}_C$ (ours) | | **56.8** | 39.1 | **48.7** | 34.1 | 43.0 | 4.3 | 55.7 | 46.0 | 56.3 | 25.2 |

Table 17: **NeRF retrieval (ShapenetRender).** Extended version of Tab. 6 with added $\mathcal{L}_C$ results. The encoder is trained on ALL. Query/gallery combinations belong to their respective test sets.

| Method | k | Recall@k (%) ↑ | | | | | | |
|---|---|---|---|---|---|---|---|---|
| | | ALL/ALL | MLP/TRI | MLP/HASH | TRI/MLP | TRI/HASH | HASH/MLP | HASH/TRI |
| nf2vec (Zama Ramirez et al., 2024) | – | – | – | – | – | – | – | – |
| Cardace et al. (2024) | | – | – | – | – | – | – | – |
| RANDOM | | 0.01 | 0.03 | 0.03 | 0.03 | 0.03 | 0.03 | 0.03 |
| $\mathcal{L}_R$ (ours) | 1 | 0.00 | 1.80 | 0.43 | 4.48 | 1.67 | 1.06 | 0.38 |
| $\mathcal{L}_{R+C}$ (ours) | | **5.25** | **30.62** | **14.77** | **33.27** | **13.17** | **13.45** | **9.46** |
| $\mathcal{L}_C$ (ours) | | 2.86 | 16.41 | 6.19 | 15.93 | 4.35 | 5.39 | 3.99 |
| RANDOM | | 0.04 | 0.13 | 0.13 | 0.13 | 0.13 | 0.13 | 0.13 |
| $\mathcal{L}_R$ (ours) | 5 | 0.00 | 5.28 | 1.82 | 13.30 | 6.12 | 3.84 | 1.37 |
| $\mathcal{L}_{R+C}$ (ours) | | **17.72** | **59.37** | **37.98** | **61.24** | **33.40** | **32.09** | **25.13** |
| $\mathcal{L}_C$ (ours) | | 10.91 | 40.61 | 18.03 | 38.89 | 14.11 | 16.84 | 14.21 |
| RANDOM | | 0.08 | 0.25 | 0.25 | 0.25 | 0.25 | 0.25 | 0.25 |
| $\mathcal{L}_R$ (ours) | 10 | 0.00 | 8.19 | 3.08 | 19.24 | 9.23 | 6.09 | 2.68 |
| $\mathcal{L}_{R+C}$ (ours) | | **27.25** | **72.67** | **51.25** | **72.62** | **45.01** | **42.83** | **36.31** |
| $\mathcal{L}_C$ (ours) | | 18.03 | 53.75 | 26.93 | 53.63 | 22.05 | 26.30 | 21.95 |

classification results when the classifier $\mathcal{C}$ is tested on architectures seen during training, whereas Tab. 16 shows classification results when $\mathcal{C}$ is tested on architectures belonging to the families seen at training time but with different hyperparameters, which we refer to as *unseen* architectures in Sec. 4. In addition to the observations discussed in Sec. 4.2, Tabs. 15 and 16 show that $\mathcal{L}_C$ prevails over $\mathcal{L}_R$ and $\mathcal{L}_{R+C}$ in very few cases; this behavior is consistent with the less noticeable separation between class-level clusters produced by $\mathcal{L}_C$ shown in Fig. 3 and discussed in Sec. 4.1. Notably, $\mathcal{L}_C$ in the best performing method when $\mathcal{C}$ is trained on HASH and tested on TRI in Tab. 15, where $\mathcal{L}_{R+C}$ struggles much more than in all other cases where $\mathcal{C}$ is tested on architectures that are not included in its training set (i.e. white cells).

**NeRF retrieval.** Tabs. 17 and 18 show retrieval results including those produced by $\mathcal{L}_C$, which are omitted in Tabs. 6 and 7. The recall@$k$ of the RANDOM baseline is computed as

$$100 \cdot \sum_{i=1}^{k} \frac{1}{|\text{gallery}|} = \frac{100 \cdot k}{|\text{gallery}|} \qquad (5)$$

Since retrieval is an instance-level task, the contrastive loss enables $\mathcal{L}_C$ to outperform $\mathcal{L}_R$ for every query/gallery combination, albeit not as effectively as $\mathcal{L}_{R+C}$, which consistently yields the best results. In particular, when both the query and the gallery belong to ALL, $\mathcal{L}_R$ is unable to perform the task: the latent space organization shown in Fig. 3 (left) and discussed in Sec. 4.1 prevents $\mathcal{L}_R$ from being capable of recognizing the same object represented by NeRFs parameterized by different architectures. Furthermore, it is worth noting that the recall@$k$ in the ALL/ALL case is lower than in the other query/gallery combinations because the gallery is larger and the probability of missing the target instance is thus higher.

Table 18: **NeRF retrieval (Objaverse generalization).** Extended version of Tab. 7 with added $\mathcal{L}_C$ results.. The encoder is trained on ALL. Query/gallery combinations belong to their respective test sets.

| Method | $k$ | Recall@$k$ (%) ↑ | | | | | | |
|---|---|---|---|---|---|---|---|---|
| | | ALL/ALL | MLP/TRI | MLP/HASH | TRI/MLP | TRI/HASH | HASH/MLP | HASH/TRI |
| nf2vec (Zama Ramirez et al., 2024) | – | – | – | – | – | – | – | – |
| Cardace et al. (2024) | | – | – | – | – | – | – | – |
| RANDOM | | 0.02 | 0.05 | 0.05 | 0.05 | 0.05 | 0.05 | 0.05 |
| $\mathcal{L}_R$ (ours) | 1 | 0.00 | 2.60 | 0.48 | 2.71 | 0.48 | 0.74 | 0.32 |
| $\mathcal{L}_{R+C}$ (ours) | | **0.64** | **9.82** | **6.10** | **8.33** | **1.75** | **6.74** | **2.92** |
| $\mathcal{L}_C$ (ours) | | 0.18 | 3.66 | 2.23 | 3.66 | 0.96 | 3.13 | 1.80 |
| RANDOM | | 0.09 | 0.27 | 0.27 | 0.27 | 0.27 | 0.27 | 0.27 |
| $\mathcal{L}_R$ (ours) | 5 | 0.00 | 6.37 | 1.11 | 7.27 | 1.91 | 2.92 | 0.69 |
| $\mathcal{L}_{R+C}$ (ours) | | **2.69** | **23.67** | **17.94** | **20.70** | **5.73** | **19.59** | **8.49** |
| $\mathcal{L}_C$ (ours) | | 1.65 | 13.22 | 8.39 | 10.08 | 3.18 | 10.46 | 5.47 |
| RANDOM | | 0.18 | 0.53 | 0.53 | 0.53 | 0.53 | 0.53 | 0.53 |
| $\mathcal{L}_R$ (ours) | 10 | 0.00 | 9.66 | 2.39 | 10.99 | 3.24 | 4.88 | 1.80 |
| $\mathcal{L}_{R+C}$ (ours) | | **4.87** | **33.65** | **26.38** | **27.55** | **9.50** | **28.13** | **12.74** |
| $\mathcal{L}_C$ (ours) | | 3.01 | 19.53 | 13.48 | 15.29 | 5.57 | 15.45 | 8.39 |

## D IMPLEMENTATION AND HARDWARE

Our framework implementation is built upon the codebases by Zama Ramirez et al. (2024) (for the decoder and training) and Lim et al. (2024) (for the GMN architecture and graph conversion). NeRFs belonging to HASH, TRI$^{OB}$, HASH$^{OB}$, and the unseen architectures were trained with the NerfAcc framework (Li et al., 2022), as done in Zama Ramirez et al. (2024) and Cardace et al. (2024) for MLP and TRI. All our experiments were performed on a single NVIDIA RTX A6000. Training either $\mathcal{L}_R$ or $\mathcal{L}_{R+C}$ took ∼2 weeks, whereas training $\mathcal{L}_C$ took ∼4 days.

## E ADDITIONAL QUALITATIVE RESULTS

Figs. 6 and 7 show NeRF retrieval qualitative results produced by $\mathcal{L}_{R+C}$ for varying query/gallery combinations belonging to MLP, TRI, or HASH. The similarity in color and/or appearance between the first 10 nearest neighbors and the query suggests that our framework encodes both relevant information about the appearance of the object represented by the NeRF and some invariance to the architecture used to parameterize it.

Fig. 8 shows NeRF retrieval qualitative results produced by $\mathcal{L}_{R+C}$ when the query belongs to MLP$^{OB}$ (Amaduzzi et al., 2025) and the gallery belongs to MLP, TRI, or HASH. As a reminder, MLP$^{OB}$'s NeRFs are vanilla MLPs with the same architecture as those in MLP (see Tab. 14). Since our framework was trained on ShapenetRender (Xu et al., 2019) NeRFs only, the purpose of this experiment is to shed some light on its ability to generalize to unseen datasets. The 10 nearest neighbors of the MLP$^{OB}$ query reveal how remarkably well our method can retrieve NeRFs belonging to the same class as the query when the latter belongs to ShapenetRender classes (i.e. *chair*, *lamp*, and *speaker*), while mostly failing for the television query (whose closest ShapenetRender class would be *display*), although, interestingly, at least one object featuring a display with a pattern similar to that of the query is always found among the closest neighbors.

Fig. 9 compares retrieval qualitative results of $\mathcal{L}_R$ and $\mathcal{L}_{R+C}$ for one MLP$^{OB}$ (Amaduzzi et al., 2025) query. As one may expect, the contrastive term in $\mathcal{L}_{R+C}$ allows retrieving objects belonging to the same class as the query even when the gallery NeRFs are parameterized by different architectures than the query (i.e. TRI and HASH), whereas $\mathcal{L}_R$ produces reasonable results for all 10 nearest neighbors only when the gallery architecture is the same as the query (i.e. MLP), which, once again, demonstrates the importance of the contrastive objective when dealing with multiple architectures.

## F ADDITIONAL NERF CAPTIONING AND Q&A RESULTS

Tabs. 19 to 24 show extended results for the language tasks described in Sec. 4.4 by adding BLEU-1 (Papineni et al., 2002), ROUGE-L (Chin-Yew, 2004), and METEOR (Banerjee & Lavie, 2005) metrics, which are traditional handcrafted similarity measures based on n-gram statistics. As discussed

Table 19: **NeRF brief captioning (multi-architecture)**. Extended version of Tab. 8.

| Method | Test Dataset | S-BERT ↑ | SimCSE ↑ | BLEU-1 ↑ | ROUGE-L ↑ | METEOR ↑ |
|---|---|---|---|---|---|---|
| LLaNA (Amaduzzi et al., 2024) | – | – | – | – | – | – |
| $\mathcal{L}_{R+C}$ (ours) | MLP | **67.1** | **68.9** | **32.7** | **35.1** | **36.1** |
| | TRI | 66.4 | 67.8 | 32.5 | 34.4 | 35.9 |
| | HASH | 66.3 | 67.9 | 32.4 | 34.5 | 35.6 |

Table 20: **NeRF detailed captioning (multi-architecture)**. Extended version of Tab. 9.

| Method | Test Dataset | S-BERT ↑ | SimCSE ↑ | BLEU-1 ↑ | ROUGE-L ↑ | METEOR ↑ |
|---|---|---|---|---|---|---|
| LLaNA (Amaduzzi et al., 2024) | – | – | – | – | – | – |
| $\mathcal{L}_{R+C}$ (ours) | MLP | **73.3** | **75.1** | **19.7** | **32.1** | **20.1** |
| | TRI | 72.2 | 73.8 | 19.5 | 31.4 | 19.7 |
| | HASH | 72.4 | 74.0 | 19.6 | 31.7 | 19.9 |

Table 21: **NeRF single-round Q&A (multi-architecture)**. Extended version of Tab. 10.

| Method | Test Dataset | S-BERT ↑ | SimCSE ↑ | BLEU-1 ↑ | ROUGE-L ↑ | METEOR ↑ |
|---|---|---|---|---|---|---|
| LLaNA (Amaduzzi et al., 2024) | – | – | – | – | – | – |
| $\mathcal{L}_{R+C}$ (ours) | MLP | **80.9** | **81.4** | 45.6 | 52.6 | 49.5 |
| | TRI | 80.7 | 81.3 | **45.7** | **52.9** | **49.8** |
| | HASH | 80.6 | 81.3 | 45.3 | 52.4 | 49.3 |

Table 22: **NeRF brief captioning (single-architecture)**. Extended version of Tab. 11.

| Method | S-BERT ↑ | SimCSE ↑ | BLEU-1 ↑ | ROUGE-L ↑ | METEOR ↑ |
|---|---|---|---|---|---|
| LLaNA (Amaduzzi et al., 2024) | **68.6** | **70.5** | 20.6 | 28.3 | 31.8 |
| $\mathcal{L}_R$ (ours) | 68.3 | 70.2 | **33.1** | **35.3** | **36.6** |

Table 23: **NeRF detailed captioning (single-architecture)**. Extended version of Tab. 12.

| Method | S-BERT ↑ | SimCSE ↑ | BLEU-1 ↑ | ROUGE-L ↑ | METEOR ↑ |
|---|---|---|---|---|---|
| LLaNA (Amaduzzi et al., 2024) | **77.4** | **79.8** | **41.3** | **36.2** | **32.4** |
| $\mathcal{L}_R$ (ours) | 74.1 | 76.0 | 20.1 | 32.5 | 20.6 |

Table 24: **NeRF single-round Q&A (single-architecture)**. Extended version of Tab. 13.

| Method | S-BERT ↑ | SimCSE ↑ | BLEU-1 ↑ | ROUGE-L ↑ | METEOR ↑ |
|---|---|---|---|---|---|
| LLaNA (Amaduzzi et al., 2024) | **81.0** | **81.6** | **46.2** | **53.2** | **50.2** |
| $\mathcal{L}_R$ (ours) | **81.0** | **81.6** | 45.4 | 52.5 | 49.4 |

in Sec. 4.4, S-BERT (Reimers & Gurevych, 2019) and SimCSE (Gao et al., 2021) are instead based on pre-trained networks, and generally considered more effective than n-gram-based metrics at measuring the quality of the generated text (Amaduzzi et al., 2024).

# G  LLM USAGE

LLMs were used in this paper as a tool to aid and polish writing. No significant role in research ideation, experimentation, or writing was delegated to LLMs.

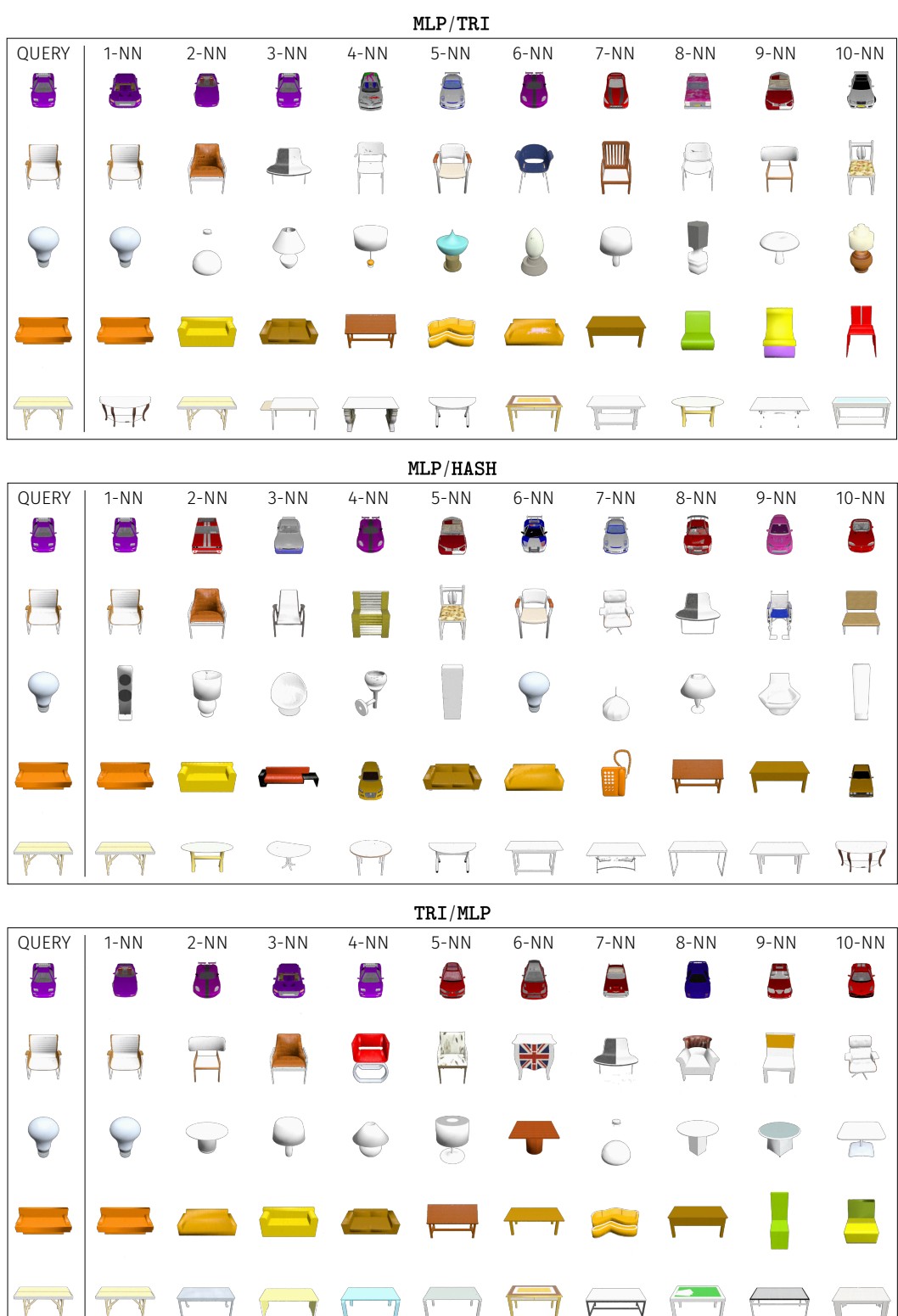

Figure 6: **NeRF retrieval qualitative results** ($\mathcal{L}_{\mathrm{R+C}}$). Query and gallery from the test set of `MLP`, `TRI`, or `HASH`.

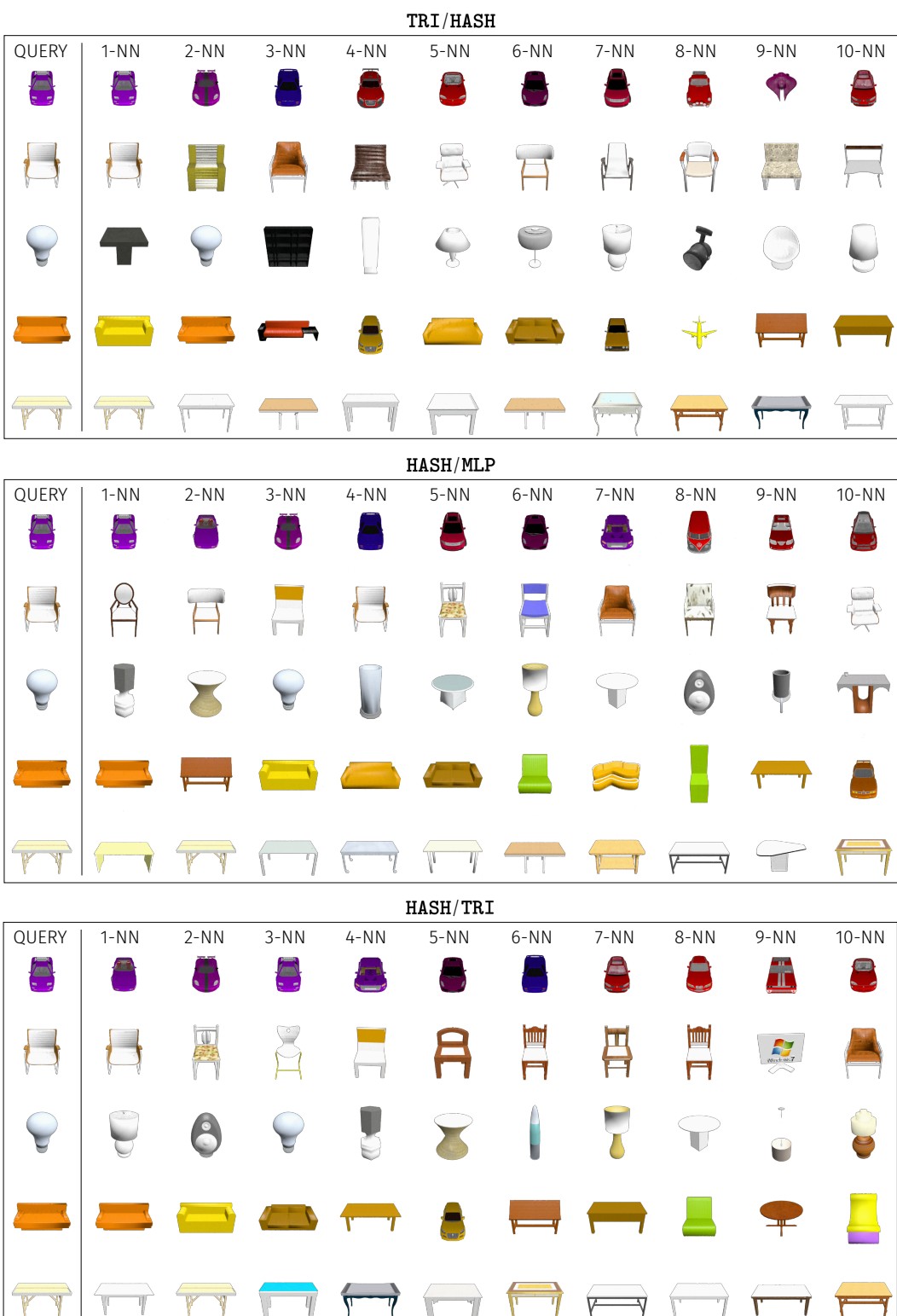

Figure 7: **NeRF retrieval qualitative results** ($\mathcal{L}_{R+C}$). Query and gallery from the test set of MLP, TRI, or HASH.

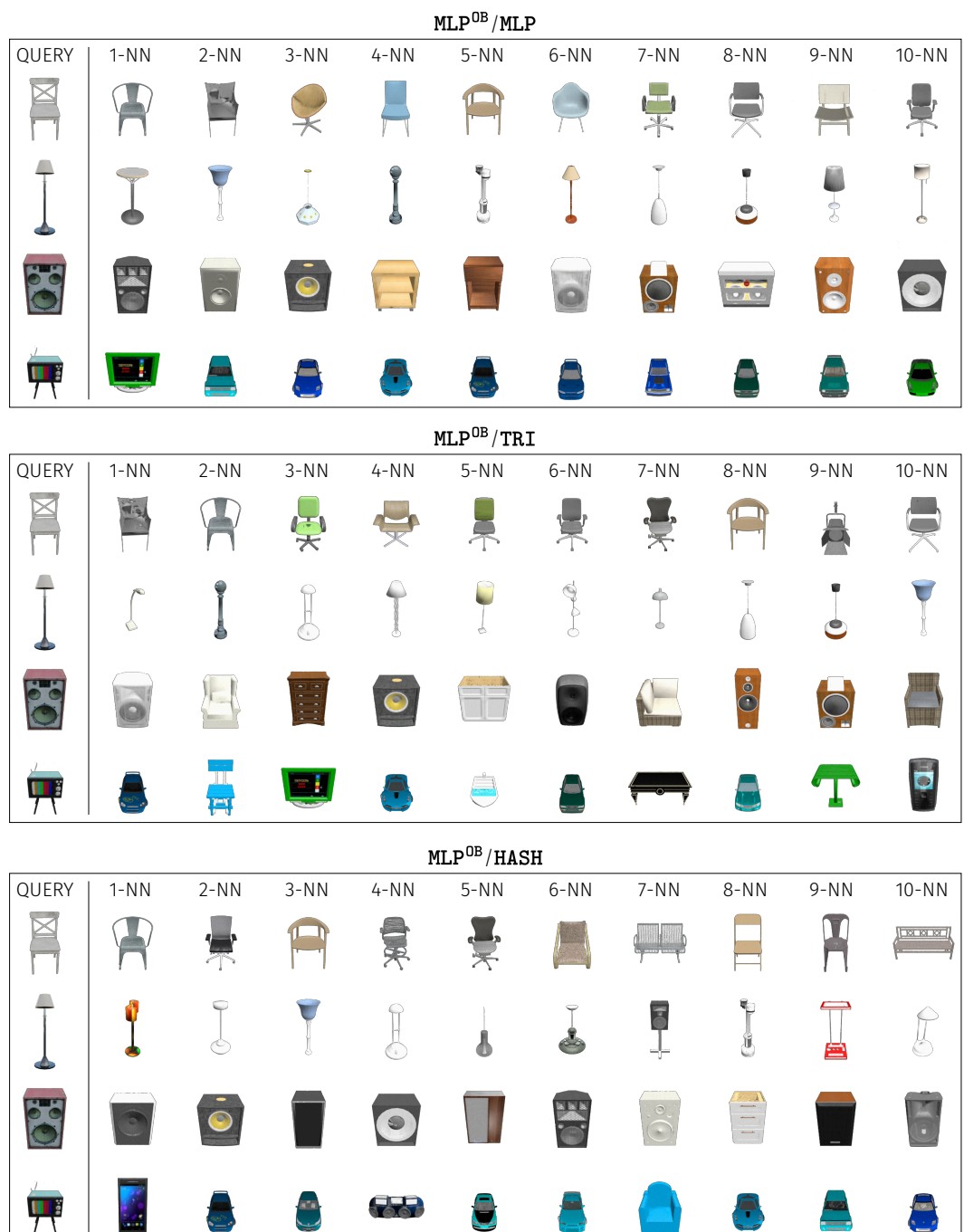

Figure 8: **Across-datasets NeRF retrieval** ($\mathcal{L}_{R+C}$). Extended version of Fig. 4. Query from the test set of MLP$^{OB}$ (Amaduzzi et al., 2025), gallery from the test set of MLP, TRI, or HASH.

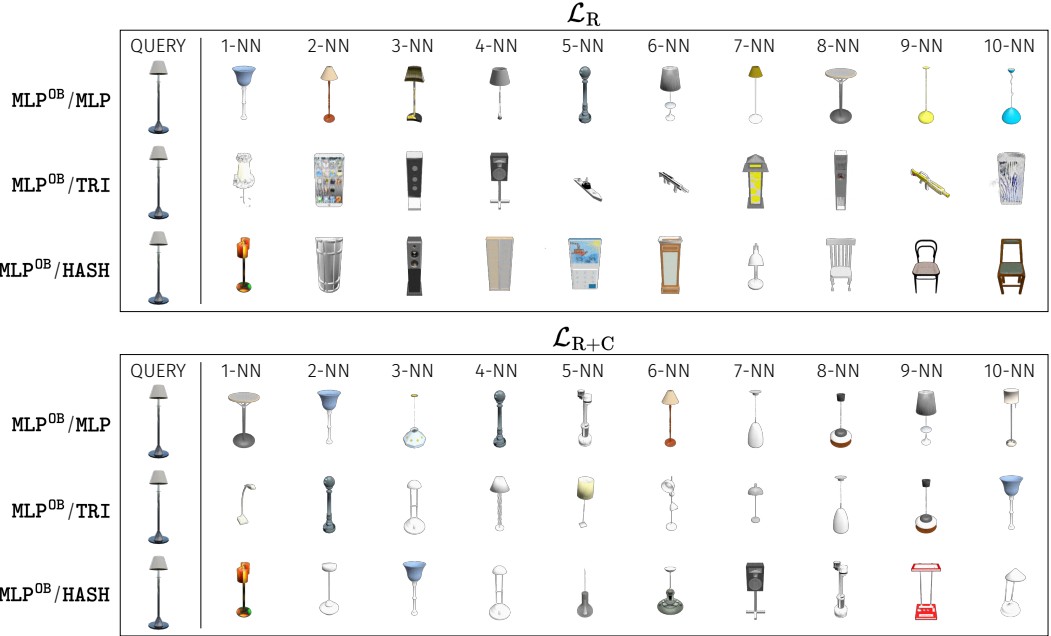

Figure 9: **Across-datasets NeRF retrieval** ($\mathcal{L}_{\text{R}}$ vs $\mathcal{L}_{\text{R+C}}$). Query from the test set of $\text{MLP}^{\text{OB}}$ (Amaduzzi et al., 2025), gallery from the test set of $\text{MLP}$, $\text{TRI}$, or $\text{HASH}$.

