# OpenReview forum: "Weight Space Representation Learning on Diverse NeRF Architectures"
_ICLR.cc/2026/Conference — ICLR 2026 Poster_

### Official Review · Reviewer_dPSz · 2025-10-30

**Soundness:** 3
**Presentation:** 3
**Contribution:** 3
**Rating:** 6
**Confidence:** 3

**Summary:**

This work addresses the challenge of learning on Neural Radiance Fields (NeRF), treating NeRF as a data format. The key question is: how can we effectively conduct learning across various NeRF formats? Given the existence of multiple NeRF formats, the learning framework varies accordingly. To accommodate these different formats—such as MLP, Triplane, and Hashgrid—this approach first converts them into a graph representation. Then, a Graph Meta-Network, equipped with an unsupervised representation learning framework, is implemented. Results indicate that employing a contrastive objective significantly enhances generalization abilities for unseen NeRF formats, as demonstrated through classification and retrieval tasks.

**Strengths:**

1. The research question is likely to attract interest from communities that utilize NeRF as a data format. The proposed technique enables direct retrieval and classification within this format.

2. The graph conversion method is novel, making it applicable to a wide range of NeRF data formats.

3. Performance results are impressive. In single architecture tasks, the method surpasses previous techniques. In multiple architecture tasks, the addition of the contrastive loss notably boosts accuracy.

**Weaknesses:**

1. While the problem is intriguing, the paper does not adequately highlight the practical uses and advantages of learning directly on various NeRF formats. In many scenarios, a single NeRF format should be sufficient for each platform. Furthermore, when it comes to classification and retrieval, the question arises: why not simply store a few images (e.g., three orthogonal views) for each NeRF and perform learning on those images? The paper mentions that rendering takes time, but this argument feels weak. Although using the complete NeRF may benefit certain dense prediction tasks, the paper does not demonstrate such complex tasks.

2. The experimental tasks performed undermines the claims made. Although the input NeRF formats are unified, the encoder architecture is limited to a Graph Network with global pooling to derive the embedding. It remains unclear how such an adaptation could extend to more complex tasks like 3D generation or 3D segmentation.

**Questions:**

Q1. Can the learned embedding be used to reconstruct the NeRF? What is the Peak Signal-to-Noise Ratio (PSNR) for the learned embedding when using the decoder?

---

> ### Author Response · Authors · 2025-11-23
> **Response to Reviewer dPSz (Part 1/3)**
>
> We thank the reviewer for acknowledging the interesting nature of our work for the wider NeRF community, its novelty, and our impressive experimental results.
>
> > **W1.1.** While the problem is intriguing, the paper does not adequately highlight the practical uses and advantages of learning directly on various NeRF formats. In many scenarios, a single NeRF format should be sufficient for each platform.
>
> As mentioned in Sec. 5, our work aims to pave the way for the development of the first foundational model for NeRFs. In this context, the ability to handle arbitrary NeRF architectures is a desirable property, as it poses no constraints on which datasets of NeRF weights the model can be trained on, thereby maximizing the amount of available training data. Furthermore, as the NeRF literature rapidly evolves, the architecture-agnostic nature of our methodology will enable it to accommodate future NeRF architectures.

---

> > ### Author Response · Authors · 2025-11-23
> > **Response to Reviewer dPSz (Part 2/3)**
> >
> > > **W1.2.** Furthermore, when it comes to classification and retrieval, the question arises: why not simply store a few images (e.g., three orthogonal views) for each NeRF and perform learning on those images? The paper mentions that rendering takes time, but this argument feels weak. Although using the complete NeRF may benefit certain dense prediction tasks, the paper does not demonstrate such complex tasks.
> >
> > Language tasks on NeRFs (e.g., NeRF captioning and Q&A) are a family of tasks for which processing NeRF weights has recently been shown to yield superior results compared to methods that ingest rendered images as input [1]. Although they are not dense tasks per se, language tasks benefit from a holistic 3D understanding of the object that is more effectively captured by the complete 3D model encoded in the NeRF weights.
> >
> > Specifically, LLaNA (Large Language and NeRF Assistant) [1] projects a NeRF embedding [2] into the input embedding space of an LLM (i.e., LLaMA 2), combines the NeRF embedding with a textual prompt, and trains the LLM to predict the text corresponding to the desired language task on the input NeRF. LLaNA's authors test their framework on MLP-based NeRFs trained on ShapenetRender (i.e., $\texttt{MLP}$), which are converted into embeddings by a frozen nf2vec encoder [2].
> >
> > We replaced the nf2vec encoder in LLaNA's pipeline with our GMN encoder, trained the resulting framework on textual descriptions from the ShapeNeRF–Text dataset [1], and evaluated it on three language tasks: brief captioning, detailed captioning, and single-round question-answering (Q&A). As in our classification experiments of Sec. 4.2, our results are divided into two categories:
> >
> > - **Multi-architecture.** Our framework ($\mathcal{L}_\text{R+C}$) is trained on $\texttt{ALL}$ (i.e., $\texttt{MLP}\cup\texttt{TRI}\cup\texttt{HASH}$), and the embeddings from $\texttt{ALL}$ produced by the frozen GMN encoder are used to train the LLM. Since the nf2vec encoder can process only MLP-based NeRFs, the original LLaNA framework cannot be applied in the multi-architecture setting.
> >
> > - **Single-architecture.** Our framework ($\mathcal{L}_\text{R}$) is trained on $\texttt{MLP}$, and the embeddings from $\texttt{MLP}$ produced by the frozen GMN encoder are used to train the LLM. In this setting, our results are compared with LLaNA, which is also trained on $\texttt{MLP}$.
> >
> > These are our results, divided by setting and task, where the language similarity metrics are those used in [1]:
> >
> > **Tab. 1.** Multi-architecture brief captioning:
> >
> > Model|Test Dataset|S-BERT|SimCSE|BLEU-1|ROUGE-L|METEOR
> > --|:--:|:--:|:--:|:--:|:--:|:--:
> > LLaNA|–|–|–|–|–|–
> > $\mathcal{L}_\text{R+C}$ (ours)|$\texttt{MLP}$|67.1|68.9|32.7|35.1|36.1
> > $\mathcal{L}_\text{R+C}$ (ours)|$\texttt{TRI}$|66.4|67.8|32.5|34.4|35.9
> > $\mathcal{L}_\text{R+C}$ (ours)|$\texttt{HASH}$|66.3|67.9|32.4|34.5|35.6
> >
> > **Tab 2.** Multi-architecture detailed captioning:
> >
> > Model|Test Dataset|S-BERT|SimCSE|BLEU-1|ROUGE-L|METEOR
> > --|:--:|:--:|:--:|:--:|:--:|:--:
> > LLaNA|–|–|–|–|–|–
> > $\mathcal{L}_\text{R+C}$ (ours)|$\texttt{MLP}$|73.3|75.1|19.7|32.1|20.1
> > $\mathcal{L}_\text{R+C}$ (ours)|$\texttt{TRI}$|72.2|73.8|19.5|31.4|19.7
> > $\mathcal{L}_\text{R+C}$ (ours)|$\texttt{HASH}$|72.4|74.0|19.6|31.7|19.9
> >
> > **Tab 3.** Multi-architecture single-round Q&A:
> >
> > Model|Test Dataset|S-BERT|SimCSE|BLEU-1|ROUGE-L|METEOR
> > --|:--:|:--:|:--:|:--:|:--:|:--:
> > LLaNA|–|–|–|–|–|–
> > $\mathcal{L}_\text{R+C}$ (ours)|$\texttt{MLP}$|80.9|81.4|45.6|52.6|49.5
> > $\mathcal{L}_\text{R+C}$ (ours)|$\texttt{TRI}$|80.7|81.3|45.7|52.9|49.8
> > $\mathcal{L}_\text{R+C}$ (ours)|$\texttt{HASH}$|80.6|81.3|45.3|52.4|49.3
> >
> > **Tab 4.** Single-architecture brief captioning:
> >
> > Model|S-BERT|SimCSE|BLEU-1|ROUGE-L|METEOR
> > --|:--:|:--:|:--:|:--:|:--:
> > LLaNA|68.6|70.5|20.6|28.3|31.8
> > $\mathcal{L}_\text{R}$ (ours)|68.3|70.2|33.1|35.3|36.6
> >
> > **Tab 5.** Single-architecture detailed captioning:
> >
> > Model|S-BERT|SimCSE|BLEU-1|ROUGE-L|METEOR
> > --|:--:|:--:|:--:|:--:|:--:
> > LLaNA|77.4|79.8|41.3|36.2|32.4
> > $\mathcal{L}_\text{R}$ (ours)|74.1|76.0|20.1|32.5|20.6
> >
> > **Tab 6.** Single-architecture single-round Q&A:
> >
> > Model|S-BERT|SimCSE|BLEU-1|ROUGE-L|METEOR
> > --|:--:|:--:|:--:|:--:|:--:
> > LLaNA|81.0|81.6|46.2|53.2|50.2
> > $\mathcal{L}_\text{R}$ (ours)|81.0|81.6|45.4|52.5|49.4
> >
> > Two main conclusions can be drawn from the tables above:
> >
> > - In the multi-architecture setting, our method is remarkably robust to the choice of NeRF architecture.
> > - In the single architecture setting, our method is, for the most part, comparable with LLaNA, especially when focusing on metrics based on pretrained encoders (i.e., S-BERT and SimCSE), which are generally more faithful in measuring text quality than n-gram-based handcrafted metrics (i.e., BLEU-1, ROUGE-L, and METEOR).
> >
> > We sincerely thank the reviewer for prompting us to run this experiment, which significantly strengthened the empirical validation of our work. We will add these results to the revised version of the manuscript.

---

> > > ### Author Response · Authors · 2025-11-23
> > > **Response to Reviewer dPSz (Part 3/3)**
> > >
> > > > **W2.** The experimental tasks performed undermines the claims made. Although the input NeRF formats are unified, the encoder architecture is limited to a Graph Network with global pooling to derive the embedding. It remains unclear how such an adaptation could extend to more complex tasks like 3D generation or 3D segmentation.
> > >
> > > Previous work in neural field processing [2] provides empirical evidence for the effectiveness of global embeddings in 3D segmentation and 3D generation tasks, while focusing on a single neural architecture (i.e., MLPs). This is achieved by using the embeddings to condition a segmentation decoder (for 3D segmentation) or to train a LatentGAN (for 3D generation). Similarly, the embeddings produced by our encoder could be exploited in such scenarios, with the added benefit of overcoming the single-architecture constraint.
> > >
> > > However, since the primary goal of our work is to introduce, for the first time, multiple NeRF architectures within a representation-learning framework operating in weight space, most of our experimental efforts were dedicated to addressing the multi-architecture nature of the problem. As a result, we chose to focus on classification and retrieval tasks to validate the downstream effectiveness of our method.
> > >
> > > Nonetheless, we agree with the reviewer that extending our empirical evaluation to more complex tasks, such as 3D segmentation and 3D generation, is a compelling research direction for future work.
> > >
> > > > **Q1.** Can the learned embedding be used to reconstruct the NeRF? What is the Peak Signal-to-Noise Ratio (PSNR) for the learned embedding when using the decoder?
> > >
> > > Yes, the embedding can be used to reconstruct views of the 3D object that the NeRF was trained on via the decoder. Indeed, the rendering loss aims at minimizing the difference between the RGB values reconstructed from the NeRF and those reconstructed from the decoder (see Eq. 1)—and, for this reason, it could equivalently be called *reconstruction loss*.
> > >
> > > The following table shows the average PSNR computed on the test set of $\texttt{MLP}$, $\texttt{TRI}$, and $\texttt{HASH}$ when performing reconstruction from the embeddings with the decoder trained with rendering-only loss (row 1) or the combination of rendering and contrastive losses (row 2). The better class separation achieved by the rendering-only objective, as shown in Fig. 3, is reflected in a higher reconstruction quality when compared with $\mathcal{L}_\text{R+C}$. The PSNR obtained when reconstructing views from the NeRF itself (row 3) should be considered an upper bound, since the RGB values reconstructed from the NeRF are what the decoder is trained to approximate.
> > >
> > > ||$\texttt{MLP}$|$\texttt{TRI}$|$\texttt{HASH}$|
> > > |--|--:|--:|--:|
> > > |Embedding + Decoder ($\mathcal{L}_\text{R}$)|19.0|18.3|18.8|
> > > |Embedding + Decoder ($\mathcal{L}_\text{R+C}$)|18.1|17.2|17.7|
> > > |NeRF (oracle) |37.8|38.3|38.1|
> > >
> > > **References**
> > >
> > > [1] Amaduzzi et al. LLaNA: Large Language and NeRF Assistant. NeurIPS 2024.
> > >
> > > [2] Zama Ramirez et al. Deep Learning on Object-centric 3D Neural Fields. TPAMI 2024.

---

### Official Review · Reviewer_5Dgh · 2025-10-31

**Soundness:** 3
**Presentation:** 3
**Contribution:** 2
**Rating:** 4
**Confidence:** 3

**Summary:**

The paper proposes a method for encoding different NeRF architectures to embeddings for the purpose of classification and retrieval.

In terms of architectures, MLP, triplane, and hash table-based NeRF architectures are being used as inputs for encoding.
In terms of loss, the rendering and the contrastive loss are being used.

For the encoding and decoding, a GMN-like and a nf2vec-like encoder and decoder are being used, respectively.

ShapenetRenderer and Objaverse datasets are being used for evaluation.

The authors noted the first time a hash table based spatial grid can be processed.

**Strengths:**

1. The paper is well written, the explanations are thorough, and the figures are clear. The equations (for the loss) are clearly written. And the figures demonstrating the graph representation (of MLP, triplane and hashtable), though do require background reading, but is understandable once with the proper context.
2. Supporting the MLP, triplane and hash table-based NeRFs cover most of the NeRF representation, if not all. This makes it a relatively general method for its own purpose.
3. Both the quantitative and qualitative evaluations of the method itself are thorough. The retrieval does demonstrate similar content is being retrieved.

**Weaknesses:**

1. All of the losses and the encoder/decoder (GMN/nf2vec) used in this work are existing components. Making the novelty in those domains minimal (though the author didn't claim particular novelty in that domain and faithfully discussed the original works).
2. The authors discussed its application to hash table. However, I have particular concern for this, both in terms of the novelty and soundness. Which I describe further in the question.
3. The results are mostly compared within the works' own variants with different losses. With minimal comparison with existing works, even for the MLP or triplane-based solutions.

**Questions:**

1. Does the author have additional discussion or ideas in losses/encoder/decoder?
2. hash table + GMN soundness: GMN is invariant/equivariant to standard neural network node permutation. Further, it respects the spatial content of a spatial grid (tri-plane like). Both these features are what the authors leveraged for the MLP and triplane-based NeRF in this current work. However, using a slightly different prime number for the hash function will lead to a completely different set of hash table features. I think this makes the hash table features inherently unsuitable for processing with the GMN. What's the authors' opinion on this? What features are stable, and what are unstable?
3. hash table + GMN novelty: If I didn't miss a particular major point, this component is a major one in the pipeline that didn't directly reuse existing works' components. However, I think the hash table + GMN combination maybe inherently insufficient. Or at least warrant further analysis, improvement or modifications which the authors didn't mention. Does the authors have further points on why hash table can be combined with GMN this way, what components may require more careful analysis?

**Details Of Ethics Concerns:**

No ethics concerns.

---

> ### Author Response · Authors · 2025-11-23
> **Response to Reviewer 5Dgh (Part 1/3)**
>
> We thank the reviewer for acknowledging the clarity and thoroughness of our writing and experiments, as well as the broad spectrum of NeRF representations we covered.
>
> > **W1.** All of the losses and the encoder/decoder (GMN/nf2vec) used in this work are existing components. Making the novelty in those domains minimal (though the author didn't claim particular novelty in that domain and faithfully discussed the original works).
>
> While building upon existing components, our paper investigates a research path that is, in fact, completely unexplored, i.e., performing tasks on NeRFs parameterized by different architectures by processing their weights as input, thereby offering a novel contribution to the weight-space learning domain. Furthermore, although contrastive objectives have been proposed and used in the past, we are the first to apply the contrastive learning framework to neural architectures, where positives are neural fields representing the same object but parameterized by different architectures, and negatives are neural fields representing different objects. This approach leverages the multi-architecture nature of the problem to enlarge the scope of traditional contrastive learning methods by integrating them into the growing research field of neural networks as a new data modality.
>
> > **W3.** The results are mostly compared within the works' own variants with different losses. With minimal comparison with existing works, even for the MLP or triplane-based solutions.
>
> As stated in Sec. 4, nf2vec [1] and Cardace et al. [2] are, to the best of our knowledge, the only existing methods that perform tasks on NeRFs by processing their weights as input in the single-architecture setting; hence, they are the ones being compared with our framework when trained on MLPs and tri-planes only, respectively. Conversely, when dealing with multiple architectures at once, our method is the first to be able to process NeRF weights in that setting; therefore, only comparisons with its own loss variants are possible.

---

> > ### Author Response · Authors · 2025-11-23
> > **Response to Reviewer 5Dgh (Part 2/3)**
> >
> > > **Q1.** Does the author have additional discussion or ideas in losses/encoder/decoder?
> >
> > Our method relies on a novel interpretation of the contrastive learning framework (as discussed in the answer to W1) and is not bound to any specific contrastive loss. We chose SigLIP because it has been shown to perform better than softmax-based alternatives for (relatively) small batch sizes and does not require tuning loss-specific hyperparameters [3]. However, alternatives such as InfoNCE [4] could also be used as training objectives in our framework.
> >
> > We chose a graph meta-network [5] as our framework's encoder because of its built-in equivariance to permutation symmetries occurring in input neural networks. Recently, meta-networks that encode other kinds of symmetries, such as scaling symmetries, have been proposed [6], which could provide an interesting path for future work on weight-space representation learning.
> >
> > The decoder we chose allows implementing a reconstruction loss that enforces latent vectors encoding global information about the object's appearance, which is conducive to tasks such as classification and retrieval. More complex, dense tasks that rely on local information, such as part segmentation, would require an ad hoc decoder, as done in [1].
> >
> > To further showcase the flexibility of our framework, results obtained when applying it to language-related tasks (i.e., captioning and Q&A) are shown in our response to **Reviewer dPSz** (W1.2).
> >
> > > **Q2.** hash table + GMN soundness: GMN is invariant/equivariant to standard neural network node permutation. Further, it respects the spatial content of a spatial grid (tri-plane like). Both these features are what the authors leveraged for the MLP and triplane-based NeRF in this current work. However, using a slightly different prime number for the hash function will lead to a completely different set of hash table features. I think this makes the hash table features inherently unsuitable for processing with the GMN. What's the authors' opinion on this? What features are stable, and what are unstable?
> >
> > A multi-resolution hash table consists of multiple hash tables, each storing feature vectors corresponding to the vertices of a (voxel) grid at a given resolution, from coarsest to finest. At coarser resolutions, there is a one-to-one correspondence between vertices and table entries, i.e., each vertex is mapped to its own unique feature vector in the table. Therefore, the only way that two hash tables at coarse resolution trained on the same data but with different random seeds can differ is in the ordering of the entries of the feature vectors. In other words, the only effects that training randomness can induce are permutations in the feature vectors. Since the GMN encoder is a graph neural network with a final pooling layer, it is, by design, invariant to node permutations in the input graph. Furthermore, our graph representation of a multi-resolution hash table has one node for each feature vector entry (see Fig. 5). Therefore, the GMN encoder is, by design, invariant to feature vector permutations induced by training randomness at coarser resolutions.
> >
> > At finer resolutions, multiple vertices can instead be mapped to the same table entry by a hash function $f\colon\mathbb{N}^d\to\lbrace 0\dots T-1 \rbrace$ that depends on $d$ prime numbers $\pi_1\dots\pi_d$, where $T$ is the max table size and $d=3$ for NeRFs. In this case, there are no theoretical guarantees on the robustness of GMNs to the feature vector behavior induced by the hash function. However, in practice, this is not an issue, since the multi-resolution hash table formulation by Müller et al. [7] sets the prime numbers used in the hash function to fixed values, i.e., $\pi_1=1$, $\pi_2=2 654 435 761$, and $\pi_3=805 459 861$. As a result, neither their original codebase (contained in the Tiny CUDA library) nor the NerfAcc framework [8] (built on top of Tiny CUDA) allows users to set the $\pi_i$s to different values by exposing them through its API. Therefore, our hash-based NeRFs were trained with the aforementioned fixed values of $\pi_1$, $\pi_2$, and $\pi_3$, as required by NerfAcc and according to Müller et al.'s original implementation.

---

> > > ### Author Response · Authors · 2025-11-23
> > > **Response to Reviewer 5Dgh (Part 3/3)**
> > >
> > > > **Q3.** hash table + GMN novelty: If I didn't miss a particular major point, this component is a major one in the pipeline that didn't directly reuse existing works' components. However, I think the hash table + GMN combination maybe inherently insufficient. Or at least warrant further analysis, improvement or modifications which the authors didn't mention. Does the authors have further points on why hash table can be combined with GMN this way, what components may require more careful analysis?
> > >
> > > The design of a graph representation for multi-resolution hash tables is indeed one of the contributions of our paper, which allowed us to be the first to perform tasks on NeRFs parameterized by hash tables by processing their weights as input.
> > >
> > > In order for a hash table to be processed by a GMN, it must be converted into a graph. As detailed in App. A, the motivation for our graph representation of hash tables is its higher memory efficiency compared to its naive voxel-grid-based alternative. Consider a multi-resolution hash table with $N$ levels, max table size $T$, $F$ features per entry, and max resolution $R$. Drawing an analogy with the tri-planar graph representation and explicitly modeling the underlying voxel grids with one node per grid vertex yields a graph with $\mathcal{O}(N(R^3+F))$ nodes and $\mathcal{O}(NR^3F)$ edges. Conversely, our graph construction requires $N(T+F)$ nodes and $NTF$ edges (see Fig. 5). Since the purpose of hash grids is to reduce the memory footprint by mapping multiple voxel grid locations to the same table entry (via the hashing function), $T$ should always be much smaller than $R^3$ (e.g., NeRFs in $\texttt{HASH}$ have $T=2^{12}$ vs $R^3=2^{21}$), hence $N(T+F)\ll N(R^3+F)$ and $NTF\ll NR^3F$.
> > >
> > > If we left any of the reviewers' concerns unaddressed, we kindly ask them to elaborate on which specific aspect of our combination of hash-table graph representation and GMN processing they would like us to address, so that we can discuss them in more depth.
> > >
> > > **References**
> > >
> > > [1] Zama Ramirez et al. Deep Learning on Object-centric 3D Neural Fields. TPAMI 2024.
> > >
> > > [2] Cardace et al. Neural Processing of Tri-Plane Hybrid Neural Fields. ICLR 2024.
> > >
> > > [3] Zhai et al. Sigmoid Loss for Language Image Pre-Training. ICCV 2023.
> > >
> > > [4] van den Oord et al. Representation Learning with Contrastive Predictive Coding. arXiv:1807.03748 (2018).
> > >
> > > [5] Lim et al. Graph Metanetworks for Processing Diverse Neural Architectures. ICLR 2024.
> > >
> > > [6] Kalogeropoulos et al. Scale Equivariant Graph Metanetworks. NeurIPS 2024.
> > >
> > > [7] Müller et al. Instant Neural Graphics Primitives with a Multiresolution Hash Encoding. SIGGRAPH 2022.
> > >
> > > [8] Li et al. NerfAcc: A General NeRF Acceleration Toolbox. arXiv:2210.04847 (2022).

---

### Official Review · Reviewer_X3pp · 2025-10-31

**Soundness:** 2
**Presentation:** 3
**Contribution:** 2
**Rating:** 4
**Confidence:** 3

**Summary:**

This work proposes a new framework aiming at performing downstream tasks on NeRF weights with diverse architectures. The authors claim that the proposed model can handle multiple NeRF-related architectures, including multi-resolution hash tables, and validate the effectiveness of their proposed components.

**Strengths:**

1.	This paper is well-written.
2.	It focuses on a commonly overlooked problem regarding downstream tasks on trained NeRFs.
3.	The proposed framework can handle various NeRF-related models, especially multi-resolution hash tables.

**Weaknesses:**

1.	The paper needs to provide a sufficient description of experimental details, such as training and testing datasets, as well as benchmarks.
2.	The paper claims to be able to classify different NeRF architectures, but all experiments focus on object-centric NeRFs, lacking discussion of scene-based models.
3.	Due to incomplete experimental details, it is difficult to evaluate the effectiveness and generalizability of the proposed method.
4.	The experimental section lacks discussion of some hyperparameters. Additionally, it lacks analysis of the experimental results, especially regarding the performance differences of the proposed method across different NeRF variants.

**Questions:**

How many different categories can the proposed model handle?

---

> ### Author Response · Authors · 2025-11-23
> **Response to Reviewer X3pp**
>
> We thank the reviewer for acknowledging the quality of our paper's writing and the overlooked nature of the problem we tackled.
>
> > **W1.** The paper needs to provide a sufficient description of experimental details, such as training and testing datasets, as well as benchmarks.
>
> > **W3.** Due to incomplete experimental details, it is difficult to evaluate the effectiveness and generalizability of the proposed method.
>
> > **W4.1.** The experimental section lacks discussion of some hyperparameters.
>
> App. B contains an extensive description of the experimental details of our datasets and training hyperparameters, which we were unable to fit into the main paper due to the page limit. A condensed version of this description can be found at the beginning of Sec. 4 in the main paper (Tab. 1, in fact, is dedicated to dataset hyperparameters). Nonetheless, we are open to receiving additional feedback from the reviewer about which specific details are missing from App. B or which ones, if any, should be moved to the main paper.
>
> > **W4.2.** The experimental section [...] lacks analysis of the experimental results, especially regarding the performance differences of the proposed method across different NeRF variants.
>
> The experimental section (Sec. 4) contains thorough discussions of the experimental results of our latent space analysis (Sec. 4.1), classification experiments (Sec. 4.2), and retrieval experiments (Sec. 4.3), comparing and contrasting all the NeRF variants explored in our paper. Nevertheless, we are open to additional feedback on which specific analyses the reviewer believes should be added to Sec. 4.
>
> > **W2.** The paper claims to be able to classify different NeRF architectures, but all experiments focus on object-centric NeRFs, lacking discussion of scene-based models.
>
> Our focus on object-centric NeRFs does not invalidate our claim of being able to process diverse NeRF architectures, as the NeRF datasets featured in our paper are, in fact, both object-centric and multi-architecture.
>
> Nonetheless, we agree with the reviewer that extending our framework to scene-based NeRFs would be an interesting research direction. However, since no prior work has performed tasks on NeRF weights parameterized by different architectures, tackling scene-based NeRFs would require creating additional datasets of NeRF weights paired with labels for tasks that are well-suited to 3D scenes. This effort exceeds our computational capabilities, given the time constraints of this discussion phase.
>
> As an example of suitable tasks for an extension to scene-based NeRFs, our response to **Reviewer dPSz** (W1.2) shows results for (object-centric) NeRF captioning and NeRF Q&A.
>
> > **Q1.** How many different categories can the proposed model handle?
>
> As described in App. B, ShapenetRender [1] is a dataset of RGB images and class labels for synthetic objects from 13 classes in the ShapeNetCore dataset [2] (airplane, bench, cabinet, car, chair, display, lamp, speaker, rifle, sofa, table, phone, and watercraft), whereas Objaverse [3] is large-scale dataset of 3D objects with no class labels. NeRFs trained on ShapenetRender are featured in both our classification and retrieval experiments (Secs. 4.2 and 4.3, respectively), while NeRFs trained on Objaverse, since they are unlabeled, can be used only in our retrieval experiments.
>
> **References**
>
> [1] Xu et al. DISN: Deep Implicit Surface Network for High-quality Single-view 3D Reconstruction. NeurIPS 2019.
>
> [2] Chang et al. ShapeNet: An Information-Rich 3D Model Repository. arXiv:1512.03012 (2015).
>
> [3] Deitke et al. Objaverse: A Universe of Annotated 3D Objects. CVPR 2023.

---

### Official Review · Reviewer_bJuV · 2025-11-02

**Soundness:** 3
**Presentation:** 3
**Contribution:** 3
**Rating:** 6
**Confidence:** 3

**Summary:**

This paper proposes an architecture-agnostic framework for weight-space representation learning on NeRFs. It encodes diverse NeRF architectures into a shared latent space using a Graph Meta-Network encoder plus an nf2vec-style decoder, trained with a rendering loss and a contrastive loss to align cross-architecture embeddings. Experiments over 13 architectures show strong performance on classification and instance-level retrieval, including unseen-architecture and cross-dataset scenarios.

**Strengths:**

- The paper tackles an underexplored problem: processing NeRFs directly in weight space across heterogeneous architectures. Prior art (nf2vec; tri-plane-specific encoders) is bound to a single architecture; this work demonstrates a unified encoder that also handles hash-table NeRFs

- The study spans 13 variants across three families, with both single- and multi-architecture training, and evaluates seen vs. unseen configurations. Results include detailed classification tables and retrieval

- A unified weight-space interface for NeRFs could enable future foundation-style models for NeRF manipulation, indexing, and understanding without rendering, saving compute and simplifying pipelines.

**Weaknesses:**

- Most training/evaluation is on synthetic datasets (ShapeNetRender), with generalization probed only via limited Objaverse tests. It remains unclear how robust the encoder is to real-world NeRFs (capture noise, exposure variation, backgrounds) or to tasks where material/lighting realism matters. A larger cross-dataset study (e.g., forward-train on Objaverse; test on different captured NeRF sets) would strengthen claims.

- The paper focuses on classification and retrieval. These are important, but practical NeRF workflows also require detection/segmentation in NeRF space, quality assessment, editability guidance, or content moderation. Demonstrating that the embeddings transfer to such tasks (even with light adapters) would better justify the "general weight-space representation" claim.

**Questions:**

Under explicit weight permutations (e.g., permuting hidden units or hash entries), does the embedding remain stable? Any empirical test or theoretical discussion?

What are typical graph sizes and embedding latencies for MLP/TRI/HASH, and how do they scale with resolution, channels, and table size?

---

> ### Author Response · Authors · 2025-11-23
> **Response to Reviewer bJuV (Part 1/2)**
>
> We thank the reviewer for acknowledging the underexplored nature of the problem tackled in our work and the strong performance and level of detail showcased in our experimental results.
>
> > **W1.** Most training/evaluation is on synthetic datasets (ShapeNetRender), with generalization probed only via limited Objaverse tests. It remains unclear how robust the encoder is to real-world NeRFs (capture noise, exposure variation, backgrounds) or to tasks where material/lighting realism matters. A larger cross-dataset study (e.g., forward-train on Objaverse; test on different captured NeRF sets) would strengthen claims.
>
> Since our work is the first to perform tasks on NeRFs parameterized by different architectures by processing their weights as input, there are no other datasets of multi-architecture NeRF weights available in the literature. Therefore, experimenting with real-world, in-the-wild NeRFs would require the creation of additional datasets of multi-architecture weights. Similarly, training on Objaverse would require creating the tri-plane and hash table counterparts of its MLP-only version [1], which we did for the test set alone to evaluate the generalization capabilities of our method. Both creating datasets of real-world NeRFs and extending the Objaverse training set to include tri-planar and hash-based NeRFs are computational efforts beyond the scope of this discussion period, given our limited computational resources and the time at our disposal.
>
> However, we recognize the value of these extensions; indeed, Sec. 5 acknowledges large-scale training on Objaverse or similarly sized datasets as the main direction for future work, eventually leading to the first foundational model for NeRFs.
>
> > **W2.** The paper focuses on classification and retrieval. These are important, but practical NeRF workflows also require detection/segmentation in NeRF space, quality assessment, editability guidance, or content moderation. Demonstrating that the embeddings transfer to such tasks (even with light adapters) would better justify the "general weight-space representation" claim.
>
> The main goal (and claim) of our paper is to introduce, for the first time, multiple NeRF architectures within a representation-learning framework operating in weight space. Since most of our experimental efforts were dedicated to addressing the multi-architecture nature of the problem, we chose to focus on classification and retrieval as validation tasks for the downstream effectiveness of our method.
>
> To the best of our knowledge, previous works that perform more complex tasks on NeRFs either operate in image space (by acting on rendered views) or, if they propose weight-space methods, are only applicable to or tested in the single-architecture scenario. As a result, experimenting with tasks such as segmentation and detection would require creating ad hoc datasets of multi-architecture NeRF weights paired with the annotations needed for those tasks, which exceeds our computational capabilities given the time constraints of this discussion phase.
>
> Nevertheless, we agree with the reviewer that extending our empirical evaluation to more complex tasks is an important research direction. Indeed, results on language tasks for NeRFs (i.e., captioning and Q&A), which did not require the creation of additional datasets, are presented in our response to **Reviewer dPSz** (W1.2).

---

> > ### Author Response · Authors · 2025-11-23
> > **Response to Reviewer bJuV (Part 2/2)**
> >
> > > **Q1.** Under explicit weight permutations (e.g., permuting hidden units or hash entries), does the embedding remain stable? Any empirical test or theoretical discussion?
> >
> > A GNN producing graph-level predictions (e.g., by pooling over predicted node and/or edge features in the final layer) is, by construction, invariant to node permutations of the input graph. Our embeddings are computed by average-pooling over the edge features predicted by a GNN encoder when the input is a graph representation of a neural architecture. The MLP graph has one node for each neuron, and the hash table graph has one node for each table entry. Therefore, our embeddings are, by construction, invariant to permutations of both hidden neurons and hash entries.
> >
> > Empirical validation of this claim lies in our current experimental evaluation, as most datasets used in our experiments contain NeRFs trained with different random seeds (i.e., different initialization and mini-batch data shuffling), which are known to lead to hidden neuron permutations [2]. The only exception is the $\texttt{MLP}$ dataset, whose authors created by setting the same initialization for all NeRFs [3]. The shared initialization is, in fact, key to the nf2vec framework's convergence, as its encoder is not permutation-invariant and is therefore less robust to the training randomness of the input NeRFs. Our framework, on the other hand, can process NeRFs with different initializations thanks to the encoder's permutation invariance.
> >
> > > **Q2.** What are typical graph sizes and embedding latencies for MLP/TRI/HASH, and how do they scale with resolution, channels, and table size?
> >
> > Let $\mathcal{G}=(\mathcal{V},\mathcal{E})$ be the parameter graph of a NeRF architecture (see Fig. 5 for reference):
> >
> > - **MLP.** Let $I$, $H$, and $O$ be input, hidden, and output dimensions, respectively, and let $L$ be the number of hidden layers. The MLP graph has $|\mathcal{V}|=I+(H+1)L+O+1$ nodes and $|\mathcal{E}|=(I+(L-1)H+L+O)H+O$ edges. For MLPs in $\texttt{MLP}$, $I=3$, $H=64$, $O=4$, and $L=3$ (see Tab. 1), hence $|\mathcal{V}|=203$ and $|\mathcal{E}|=8836$.
> >
> > - **Tri-plane.** Let $W$ be the tri-plane resolution and let $C$ be the number of channels. The tri-plane graph has $|\mathcal{V}|=3W^2+C$ nodes and $|\mathcal{E}|=3W^2C$ edges, i.e., both the number of nodes and the number of edges scale linearly with the number of channels and quadratically with the resolution. For tri-planes in $\texttt{TRI}$, $W=32$ and $C=16$ (see Tab. 1), hence $|\mathcal{V}|=3088$ and $|\mathcal{E}|=49152$ (plus the nodes and edges of the MLP that follows).
> >
> > - **Hash table.** Let $N$, $T$, and $F$ be the number of resolution levels, the max table size, and the number of features per table entry, respectively. The hash table graph has $|\mathcal{V}|=N(T+F)$ nodes and $|\mathcal{E}|=NTF$ edges, i.e., both the number of nodes and the number of edges scale linearly with the table size and with the number of features per entry. For hash tables in $\texttt{HASH}$, $N=4$, $T=2^{12}$, and $F=2$ (see Tab. 1), hence $|\mathcal{V}|=16392$ and $|\mathcal{E}|=32768$ (plus the nodes and edges of the MLP that follows).
> >
> > Embeddings are 1024-dimensional vectors, regardless of the architecture they encode (see App. B).
> >
> > **References**
> >
> > [1] Amaduzzi et al. Scaling LLaNA: Advancing NeRF-Language Understanding Through Large-Scale Training. arXiv:2504.13995 (2025).
> >
> > [2] Entezari et al. The Role of Permutation Invariance in Linear Mode Connectivity of Neural Networks. ICLR 2022.
> >
> > [3] Zama Ramirez et al. Deep Learning on Object-centric 3D Neural Fields. TPAMI 2024.

---

### Author Response · Authors · 2025-11-23
**Global Response**

Dear reviewers,

Thank you for your constructive and valuable feedback. Our answers to your questions and concerns can be found in the comments below. We look forward to engaging in further discussion and are happy to address any additional questions.

---

### Comment · Area_Chair_P5JN · 2025-11-28
**Discussions**

Dear Reviewers,

The authors have submitted their rebuttals to your reviews. Could you please express your opinions?
Many thanks,
AC

---

### Author Response · Authors · 2025-12-02
**Rebuttal Summary**

We thank the reviewers again for their valuable feedback. In the following, we gather the strengths pointed out by the reviewers and summarize our responses to their questions and concerns.

**Strengths.** Reviewers acknowledged the value of our contribution across four main dimensions:

1. **Relevant research direction**
    - "The paper tackles an underexplored problem" (Reviewer bJuV).
    - "[The paper] focuses on a commonly overlooked problem" (Reviewer X3pp).
    - "The research question is likely to attract interest from communities that utilize NeRF as a data format" (Reviewer dPSz).

1. **Clear and well-written paper**
    - "This paper is well-written" (Reviewer X3pp).
    - "The paper is well written, the explanations are thorough, and the figures are clear" (Reviewer 5Dgh).
    - " The equations (for the loss) are clearly written"  (Reviewer 5Dgh).

1. **Comprehensive methodology**
    - "Supporting the MLP, triplane and hash table-based NeRFs cover most of the NeRF representation, if not all" (Reviewer 5Dgh).
    - "The graph conversion method is novel, making it applicable to a wide range of NeRF data formats" (Reviewer dPSz).

1. **Strong empirical results**
    - "Experiments over 13 architectures show strong performance on classification and instance-level retrieval, including unseen-architecture and cross-dataset scenarios" (Reviewer bJuV).
    - "Both the quantitative and qualitative evaluations of the method itself are thorough" (Reviewer 5Dgh).
    - "Performance results are impressive" (Reviewer dPSz).

**Rebuttal.** Reviewers' comments shared two main common threads: concerns about the interplay between the scalability of our graph representation for multi-resolution hash tables and the robustness of our framework when processing such representation (Reviewers bJuV and 5Dgh), and the lack of more complex tasks beyond classification and retrieval to validate the downstream effectiveness of our representation learning framework (Reviewers bJuV and dPSz):

- **Scalability of our graph representation of hash tables.** In the answer to Reviewer bJuV (Q2), we analytically show that both the number of nodes and the number of edges of our graph representation of a multi-res hash table scale *linearly* with the table size and with the number of features per entry, thereby providing a memory-efficient alternative to the naive voxel-based graph representation, which instead scales cubically with the resolution, as discussed in the answer to Reviewer 5Dgh (Q3).

- **Permutation invariance of our framework.** In the answer to Reviewer bJuV (Q1), we discuss how our framework is, by design, *invariant* to hidden neuron permutations of input MLPs induced by training randomness, and, as a result, able to process NeRFs trained with different random seeds, unlike previous NeRF-processing methods [1]. Analogously, training randomness might induce permutations in hash table entries, which our framework is, once again, invariant to by construction, as detailed in the answer to Reviewer 5Dgh (Q2).

- **Extension to language tasks.** To address a comment from Reviewer dPSz (W1.2), we tested our framework on language-related tasks, where processing NeRF weights has previously been shown to lead to superior results compared to methods that ingest rendered images as input [2], further justifying our weight-space-based approach. Following the LLaNA framework [2], we used the embeddings produced by our trained encoder to condition an LLM for NeRF captioning and NeRF question-answering. When trained on MLP-based, tri-planar, and hash-based NeRFs together (multi-architecture), our method shows strong and consistent results across architectures. When restricted to MLP-based NeRFs, as in [2] (single-architecture), our method achieves results comparable to those of the original LLaNA framework. These results significantly strengthen our work, as they show that the embeddings produced by our representation learning framework retain both their architecture-agnostic nature and their effectiveness even for complex language tasks that require a holistic 3D understanding of the object.

We added results on language tasks, originally found in the answer to Reviewer dPSz (W1.2), to the revised paper, marking them in blue along with the other changes needed to fit this addition to the original manuscript.

**References**

[1] Zama Ramirez et al. Deep Learning on Object-centric 3D Neural Fields. TPAMI 2024.

[2] Amaduzzi et al. LLaNA: Large Language and NeRF Assistant. NeurIPS 2024.

---

### Meta-Review · Area_Chair_Mjav · 2026-01-07

**Summary:**

Recent methods demonstrated that the NeRFs weights can be used as input for frameworks designed for deep learning tasks.  The paper proposes a framework to process NeRFs with diverse architectures and do inference on unseen architectures.  The authors train a Graph Meta-Network within an unsupervised representation learning framework, and show that the contrastive objective obtains an architecture-agnostic latent space.

The paper received mixed reviews, but they all seem positive after the rebuttal from the authors.  There are still some minor issues remaining.  But, overall, the strengths of the paper outweigh the negative issues with the reused components and the limited evaluation.  Thus, I recommend the acceptance of the paper.

Strengths:
- Tackling the processing of NeRFs directly in weight space across heterogeneous architectures is an unexplored problem
- Extensive evaluation on 13 variants across 3 families
- Unified weight-space interface for NeRFs could enable future foundation-style models for NeRF manipulation, indexing, and understanding without rendering
- Supporting the MLP, tri plane, and hash table-based NeRFs cover most, if not all, of the NeRFs representations.
- The graph conversion methods is new

Weaknesses:
- It is unclear how robust the encoder is to real-world NeRFs
- The evaluation focuses on classification and retrieval, but misses other tasks like detection, segmentation, quality assessment, editability guidance, or content moderation
- The losses and the encoder/decoders are existing components
- The comparisons with existing works are minimal
- The hash table and GMN soundness and novelty was questioned
- No highlight on the practical uses and advantages of learning on various NeRF formats

**Reviewer Concerns:**

Reviewer bJuV raised concerns about the robustness of the encoder, the limited practical evaluation of the NeRF space tasks, and the stability of the embeddings.  The authors replied to the concerns and addressed the issues raised in the review.

Reviewer X3pp raised concerns about the object-centric NeRFs experiments, and lacks analysis of experimental results.  The authors replied and explained that the descriptions are in the appendix and justified that the object-centric focus doesn't invalidate the results.

Reviewer 5Dgh raised concerns about the limited technical contribution given the reuse of existing components, the minimal comparison with existing works, and the novelty and soundness of the hash table and the GMN.  The authors addressed the issues justifying that the use of existing components is not an issue given that it is exploring a new research path.  The authors also explain that the graph representations for multi-resolution hash tables is a contribution of the paper.

Reviewer dPSz raised concerns about the lack of discussion of practical advantages of learning directly on various NeRF formats, and the experimental tasks undermines the claims made.  The authors addressed the comments and explained that the development of the proposal paves the way of foundation models for NeRFs, and presented results to show that the method works on diverse tasks regardless of the NeRF architecture.

**Reviewer Scores:**

Reviewer bJuV recommended a weak accept. And most of the reviewer concerns were solved, thus, strengthning the accept recommendation.

Reviewer X3pp recommended a weak reject.  After the rebuttal, the authors address the concerns raised by the reviewer.  Thus, I consider the recommendation a weak positive one.

Reviewer 5Dgh recommended a weak reject. Given the responses by the authors I consider the evaluation of the paper a positive one.

Reviewer dPSz recommended a weak accept.  Given that the authors addressed the reviewer's concerns, I consider the recommendation a more positive one.

---

### Decision · Program_Chairs · 2026-01-26

Accept (Poster)